# Dendritic Cells in Anticancer Vaccination: Rationale for Ex Vivo Loading or In Vivo Targeting

**DOI:** 10.3390/cancers12030590

**Published:** 2020-03-05

**Authors:** Alexey V. Baldin, Lyudmila V. Savvateeva, Alexandr V. Bazhin, Andrey A. Zamyatnin

**Affiliations:** 1Institute of Molecular Medicine, Sechenov First Moscow State Medical University, 119991 Moscow, Russia; alexeyvbaldin@gmail.com (A.V.B.); ludmilaslv@yandex.ru (L.V.S.); 2Department of General, Visceral and Transplant Surgery, Ludwig-Maximilians University of Munich, 81377 Munich, Germany; alexandr.bazhin@med.uni-muenchen.de; 3German Cancer Consortium (DKTK), Partner Site Munich, 80336 Munich, Germany; 4Belozersky Institute of Physico-Chemical Biology, Department of Cell Signaling, Lomonosov Moscow State University, 119991 Moscow, Russia

**Keywords:** cancer immunotherapy, combination immunotherapy, anticancer vaccine, dendritic cells, dendritic cell vaccine, dendritic cell targeting

## Abstract

Dendritic cells (DCs) have shown great potential as a component or target in the landscape of cancer immunotherapy. Different in vivo and ex vivo strategies of DC vaccine generation with different outcomes have been proposed. Numerous clinical trials have demonstrated their efficacy and safety in cancer patients. However, there is no consensus regarding which DC-based vaccine generation method is preferable. A problem of result comparison between trials in which different DC-loading or -targeting approaches have been applied remains. The employment of different DC generation and maturation methods, antigens and administration routes from trial to trial also limits the objective comparison of DC vaccines. In the present review, we discuss different methods of DC vaccine generation. We conclude that standardized trial designs, treatment settings and outcome assessment criteria will help to determine which DC vaccine generation approach should be applied in certain cancer cases. This will result in a reduction in alternatives in the selection of preferable DC-based vaccine tactics in patient. Moreover, it has become clear that the application of a DC vaccine alone is not sufficient and combination immunotherapy with recent advances, such as immune checkpoint inhibitors, should be employed to achieve a better clinical response and outcome.

## 1. Introduction

Dendritic cells (DCs) are professional antigen-presenting cells (APCs) that possess some functions which distinguish them from other APCs. Dendritic cells are significantly more efficient at T cell stimulation and are distinguished by their ability to stimulate immunologically naive T cells. Dendritic cells can encounter and activate antigen-specific CD8+ and CD4+ T cells through major histocompatilibity complex (MHC) I-T cell receptor (TCR) and MHC II-TCR interaction, respectively [1]. Meanwhile, DCs are known to express exceptionally high levels of MHC II and co-stimulatory molecules compared to monocytes. Such features allow DCs to form multiple contacts with T cells simultaneously and provide co-stimulatory signals that result in the expansion and proliferation of a large number of T cells locally [2,3]. In addition, DCs control the induction of T cell tolerance [4]. Regulatory T (Treg) cells can also be uniquely stimulated to proliferate by DCs, enhancing their immunosuppressive capabilities [5,6]. Finally, DCs can possess innate immune functions, such as the secretion of IL-12 and type I interferons (IFNs), as well as mobilizing natural killer (NK) cells, making DCs a kind of connecting link between innate and adaptive immunity [7,8,9].

There are different impact points of the immune system on tumor cells. In terms of innate immunity, NK cells play a crucial role in cancer counteraction. Although NK cells are good at controlling tumor initiation, they are often inefficacious in progressive disease. Moreover, many phenotypes of NK cells that infiltrate progressive tumors were observed to be regulatory, pro-angiogenic and low-cytotoxic, meaning that they also possess cancer-promoting properties [10].

The transformation of malignant cells by different types of mutation during their progression makes them immunogenic for the organism. This phenomenon occurs due to the atypical protein expression encoded by mutant genes. Such aberrant proteins are foreign to the immune system. Thus, the expression of foreign proteins—tumor-associated antigens (TAAs) or tumor-specific antigens (TSAs)—by malignant cells is the mechanism that allows adaptive immune system detection and the elimination of tumor cells. There are cytotoxic T lymphocytes (CTLs) capable of antigen-specific recognition and destruction of tumor cells. Cytotoxic T lymphocytes originate from their precursors—naive CD8^+^ T cells. Unlike NK cells, CD8^+^ T cells are not universal killers. Being naive T killers, they are not capable of being cytotoxic unless they, in the process known as T cell priming, receive specific signals to activate from DCs. This process involves CD8^+^ T cell activation by the presentation of an antigen by DCs through MHC I-TCR interaction accompanied by different co-stimulatory interactions, such as B7.1-CD28, CD70-CD27 and OX40L-OX40 [11]. However, despite the lack of an ability to recognize a wide spectrum of foreign cells, activated specific CTLs can develop a much stronger cytotoxic response against tumor cells carrying a specific antigen. Additionally, there are naive CD4^+^ T cells that can be activated by DCs in a similar manner as CD8^+^ T cells, but through MHC II-TCR interaction [12]. Moreover, CD8^+^ T cells can themselves recruit naive CD4^+^ T cells by directly binding to them after the acquisition of DC membrane fragments and MHC II molecules via trogocytosis, with the subsequent formation of ternary complexes, in which CD8^+^ and CD4^+^ T cells interact with DCs and with each other [13]. After the differentiation of naive CD4^+^ T cells into T helper type 1 (Th1) cells, they contribute to the potentiation of the CTL response by the production of cytokines required for CD8^+^ T cell proliferation and differentiation, as well as by increasing DCs’ ability to recruit CD8^+^ T cells [14].

The activation and proliferation of antigen-specific CD8^+^ and CD4^+^ T cells begins with cross-presentation by DCs. In this process, DCs present the antigen epitope to naive CD8^+^ or CD4^+^ T cells through MHC I-epitope or MHC II-epitope complexes on their surface, respectively. Before being presented to a naive T cell, antigens will be captured, internalized and processed by DCs, which are other DC functions. Antigen processing by DCs is one of the important steps of antigen cross-presentation. This is critically dependent on which cell compartment the antigen will be delivered to. To achieve proper cross-presentation, exogenous antigens should be routed to endosomes or phagosomes [15]. Thus, it is important that the DC receptor that performs antigen internalization can route it to the endocytic pathway. The endocytic pathway of DCs, in turn, has a reduced proteolytic capacity compared with macrophages or neutrophils [16]. It has been shown that reduced degradation of antigen correlates with cross-presentation efficacy [16,17]. Such a feature of DCs also contributes to enhanced cross-presentation, making them superior APCs compared to others. Moreover, it has been demonstrated that antigens cross-presented more efficiently if targeted to less degradative early endosomes, than to highly degradative late endosomes [18,19]. However, antigens processed by late endosomes were more efficiently cross-presented through the MHC II pathway [18]. Another pathway of exogenous antigen processing is cytosolic. Nonetheless, it first involves antigen routing to endocytic compartments with the following export to the cytosol, where antigens are exposed to proteasomal degradation [15]. Such antigen export to the cytosol is more efficient in DCs than in macrophages, and small molecules are exported much more efficiently than larger ones [20]. Thus, some degree of degradation favors cross-presentation. Another recent study demonstrated that the antigen structure could dramatically affect cross-presentation. It has been shown that large particulate glycopolymers were routed to non-endosomal compartments after internalization where antigen processing is not performed [21]. Besides antigen trafficking and structure, antigen processing by DCs also depends on different enzyme systems that are distinct for N- and C-terminal cleavage of antigens [22]. Such variety in antigen processing affects cross-presentation efficacy and can be controlled to enhance it. However, antigen processing is beyond the scope of this review and will not be further discussed. As for the capture and internalization of antigens, for this function, DCs carry a wide spectrum of different receptors that enable the internalization of different foreign substances, including antigens. Dendritic cells, being a kind of “scavenger”, non-specifically uptake self- and non-self-antigens and subject them to processing, resulting in the presentation of epitopes from processed proteins in complex with MHC class I and II on their cell surface.

In respect to this model, it has become clear that CTLs and Th1 cells are the resulting effector cells of the adaptive anticancer immune response. Their precursor cells are inactive and are in constant search of an appropriate epitope-MHC complex on a DC surface to form a strong bond with their TCR. Thus, DCs act as a kind of immune surveillance dispatcher for other immune cells where they can receive activation signals. However, besides the ability to promote T cell cytotoxicity, DCs can produce proliferation signals for T cell tolerance, which depend on the DC subset, the current state of DC maturation, the microenvironment and tissue localization. By the internalization of self-antigens in the microenvironment with the presence of anti-inflammatory cytokines, but a lack of adjuvants, pathogen-associated molecular patterns (PAMPs), like Toll-like receptor (TLR) ligands, or cluster of differentiation 40 ligand (CD40L), DCs maintain immune homeostasis by promoting peripheral T cell tolerance [23,24]. It has been shown that T cell tolerance is driven by the interaction of PD1 and CTLA4 expressed on T cells with their ligands expressed on DCs [25,26]. Moreover, PD1-PDL1 interaction was shown to be necessary for the induction of Treg cells [27]. It has also been established that tumor cells could overexpress PDL1, contributing to the immunosuppressive tumor microenvironment and immune evasion [28]. The characterized interactions between PD1 and CTLA4 with their ligands were designated as immune checkpoints. In summary, immune checkpoints are the mechanism of the immune system that prevents uncontrollable cytotoxicity manifestation. They provide a T cell inhibitory signal, which is preventative for autoimmune reactions. However, tumors have adopted this mechanism to evade immune surveillance.

The observation of such DC functions suggested a possible immunological treatment for cancer. The approach is to deliver whatever antigen is required to be eliminated into DCs as a vaccine, which will lead to the cross-presentation and development of an antigen-specific immune response. Besides the application of an anticancer vaccine, it was proposed that immune checkpoint inhibitors (ICI) could be applied to inhibit immune tolerance signals emanating from DCs and mimicking tumor cells. It has been established that single component treatment with an anticancer vaccine or ICI could produce durable clinical responses and improved survival. However, only a minority of patients from study to study experience clinical benefits. Recently, a combinatorial approach to cancer immunotherapy was proposed [29]. Such an approach allows the full potential of immunotherapy to be realized using a rational combination of complementary immunomodulatory agents [29]. Different combinations, consisting of components that engage innate and adaptive immune responses, were tested. Such combinations usually consist of any kind of anticancer vaccine, chosen immune checkpoint inhibitor, adjuvant, cytokines and tumor-specific antibodies. In a study by Moynihan et al., four different components were tested using murine models: peptides from TSAs, IL-2 with a prolonged half-life (IL-2 modified with serum mouse albumin [30]), anti-PD-1 antibody and antibodies specific to TSAs [31]. Importantly, when combination components were examined separately, each showed only moderate efficacy, whereas the combination of all components showed the greatest effect [31]. This result demonstrates that the limitations of the single immune treatment approach would be circumvented by the combination of different immunomodulatory agents. Specifically, the usage of ICI and anticancer vaccines simultaneously would enhance the clinical response by allowing each to cover the deficiencies of the other. Clinical trials with the purpose of investigating the efficacy and safety of the combinatorial approach, including the simultaneous application of anticancer vaccine and ICI, are already being undertaken [32].

In this review, we tried to focus precisely on the antigen-specific immunogenic component of the proposed treatment scheme, which is the vaccine, the role of which is played by substances that can target and activate DCs in vivo or activated DCs themselves. Technologies for antigen delivery to DCs have been developed and improved during the last 30 years. To date, there are a variety of strategies for antigen delivery to DCs, with the use of methods that differ in substance reflected in the results. All these strategies could be divided into two groups: methods of antigen delivery to DC in vivo and in vitro. In vivo strategies can be described as in vivo DC targeting. They are based on the presence of different receptors on the surface of DCs that are potential targets for specific vaccines containing the required antigens. The administration of such DC-targeting vaccines results in antigen uptake, maturation and the activation of DCs in their natural surroundings.

After methods of ex vivo DC cultivation were developed, ex vivo strategies of antigen delivery and activation of DCs were proposed. Such technologies are based on the straight loading of required antigens or antigen encoding vectors into DCs ex vivo, accompanied by the external stimulus for DC maturation using different cytokines and ligands. The received autologous mature DCs can be administered back to the patient, resulting in CTL priming.

All of the proposed strategies are quite different in methodology. The outcome of this situation is that each approach results in a different efficacy at different resource intensiveness. Furthermore, methodological differences make comparison difficult. In this review, we present a brief description of in vivo and ex vivo approaches of antigen delivery to DCs and their activation, with a discussion of their advantages and disadvantages.

## 2. In Vivo Dendritic Cell Targeting

In vivo strategies of cancer antigen delivery to DCs are based on employing DC surface molecules and their receptor machinery. This approach implies the utilization of DC receptor ligands, adjuvants, anti-receptor antibodies and other types of substance that can accurately bind to their target on DCs, followed by their uptake or changes to the DCs’ phenotype. Accordingly, there is an opportunity to deliver cancer antigens to DCs in vivo. Specially prepared vaccines consisting of cancer antigens in complex or fused to DC-targeting substances will be delivered directly to the DCs after being administered in patients’ tissues (Figure 1).

Among all DC surface molecules, several groups of them can be highlighted as being responsible for foreign or host molecule recognition and/or for antigen uptake. Dendritic cells are an important part of the immune system due to their ability to recognize different environmental signals associated with pathological processes, or so-called PAMPs, and damage-associated molecular patterns (DAMPs). For this purpose, there are different environmental sensors—pattern recognition receptors (PRRs)—on the surface of different subsets of DCs, such as TLRs, NOD-like receptors, C-type lectin receptors (CLRs), RIG-I, MDA-5 and others [33,34]. These molecules trigger DC maturation and pro-inflammatory cytokine secretion if bound to their PAMP or DAMP ligands, such as bacterial lipopolysaccharides (LPSs), endotoxins and proteins from damaged or stressed host cells, including cancer cells, host DNA and RNA, ATP and others [35,36]. Consequently, this environmental signal sensitivity of DCs could be utilized to enhance the DC-regulated immune response. For this purpose, the PRR ligands of DCs could serve as adjuvants in the composition of DC anticancer vaccines.

Another feature of DCs as professional APCs is the ability to capture and internalize antigens alone or in complex with other host proteins. As a kind of “scavenger”, DCs carry a wide spectrum of different receptors for the uptake of all manner of antigens. This incomplete list of receptors includes CD205/DEC205, Clec9a, CD206/mannose receptor (MR), CD207/langerin, CD209/dendritic cell-specific intercellular adhesion molecule-3-grabbing non-integrin (DC-SIGN), Clec4a/dendritic cell immunoreceptor (DCIR), Fc receptors (FcRs) and so-called scavenger receptors (SRs), which include Clec8a/LOX-1, FEEL-1 and SREC-1 [37,38,39,40]. Several receptors listed are expressed specifically on distinct DC subsets localized in different tissues. Accordingly, it is possible to target a specific subset of DCs, since it has become clear that reaching a specific DC subset might enable control over the type of immune response that is produced [41].

The presence of the described receptors gives DCs the natural ability to uptake antigens through mechanisms of receptor-mediated endocytosis and phagocytosis. There is, therefore, an opportunity for researchers in terms of the in vivo delivery of cancer antigens to DCs. Nothing but the administration of a rationally designed DC-targeting anticancer vaccine into DC-rich sites of patient’s body would be enough to achieve a strong anticancer response. After vaccine administration, DCs will naturally uptake vaccine-containing antigens and interact with adjuvants that will be the signal for DC maturation, migration to the lymph nodes and initiation of T cell response [42,43]. Several strategies of in vivo DC-targeting are discussed below. The majority of receptors employed for such strategies belong to the family of CLRs and most of them were assessed in preclinical studies. Human clinical trials exploring the efficacy of in vivo DC receptor targeting have recently begun and only a few of them have been completed with published results (Table 1 and Appendix A).

### 2.1. Fc Receptor Targeting

Fc receptors are classical antigen uptake receptors. These receptors are responsible for immunoglobulin-mediated cross-presentation of antigens by DCs. There are different FcRs for each class of immunoglobulin: FcαR for IgA, FcεR for IgE, FcγR for IgG and Fcα/μR for both IgA and IgM [56]. Cancer antigens can be delivered to DCs through FcR by being in complex with an antibody. It has been demonstrated in pioneering studies that bispecific antibodies recognizing FcRs and antigen induced a humoral response in mice [57]. Antigen–antibody complexes recognizing FcRs were also shown to activate cytotoxic immunity. It has been demonstrated in mice that the ovalbumin (OVA)–anti-OVA complex induced a CD8^+^ T cell response and it showed a 10-fold higher efficacy than OVA antigen administered alone [58]. Fc receptors are expressed non-specifically on different subsets of DCs, which limits the possible application of FcR-targeting vaccines to induce a subset-specific immune response. However, another study provided interesting data suggesting that FcR targeting could be more preferable than MR targeting, which will be discussed below, at least for some cancer antigens. It was demonstrated that the PSA–anti-PSA complex was associated with the induction of both CD4^+^ and CD8^+^ T cells, while mannosylated PSA led to the induction of only CD4^+^ T cells [59]. Therefore, despite the unspecific expression of FcRs on DCs, but given their ability to induce a broad range of immune responses, FcRs are included with DC receptors for antigen uptake as rational targets for cancer antigen delivery to DCs.

### 2.2. C-type Lectin Receptor Targeting

C-type lectin receptors are natural carbohydrate-recognition receptors that are highly expressed by multiple subsets of DCs [41]. Due to their involvement in the uptake and processing of antigens for cross-presentation, this receptor family has been closely examined as a target for in vivo DC targeting vaccination. Various CLRs were proposed as a target for anticancer vaccination, with MR, DC-SIGN, and DEC205 being the most often described. Most of these receptors are specifically expressed on different human DCs subsets (Table 2). Initial strategies have been developed to produce glycosylated vaccines that are preferentially taken up by DCs expressing these CLRs [41]. However, it remains difficult to target a specific subset of DCs using sugars due to their frequent recognition by multiple CLRs. To improve antigen delivery to a specific DC subset, the utilization of antigen complexes or fusions with anti-receptor antibodies was proposed and this currently seems to be a more preferable approach to in vivo CLR targeting. The first clinical trials exploring the efficacy and safety of the in vivo CLR-targeting approach in cancer started only about a decade ago and only a few of them were completed with published results (Table 1 and Appendix A).

#### 2.2.1. Mannose Receptor Targeting

The mannose receptor was initially thought to be highly expressed on DCs and a great contribution of MR to antigen capture and internalization functions of DCs was supposed. The mannose receptor was one of the first receptors proposed as a target for in vivo DC-targeting vaccines. The natural ligands of MRs are carbohydrates (mannose, glucose, maltose, fucose) presented on the cell walls of bacteria, in yeasts and viruses, as well as certain endogenous glycoproteins [68]. Using ex vivo cultures of murine macrophages and human monocyte-derived DCs (moDCs) it was shown that after being captured by MR, antigens can be internalized through phagocytosis or endocytosis, followed by involvement in processing and cross-presentation through MHC class I and II pathways [69,70]. To target MR, mannosylation of antigen has initially been performed. Indeed, mannosylated proteins and peptides was found to have 200- to 10,000-fold higher cross-presentation efficacy through the MHC class II pathway, compared to non-mannosylated [70]. In addition, it is possible to regulate which cross-presentation pathway the mannosylated protein will undergo, depending on the chemical modification of mannan. It has been demonstrated that MUC1 antigen, if conjugated to oxidized mannan (comprising aldehydes), showed a 1000-fold more efficient cross-presentation through the MHC class I pathway, compared to MUC1 antigen being conjugated to reduced mannan (no aldehydes) [69]. However, the application of MUC1-reduced mannan resulted in the MHC class II cross-presentation pathway.

Some studies have reported strategies to target MR with different TAAs fused to the anti-MR antibody. Different groups showed promising results following in vitro studies on moDCs with this strategy using pmel17, hCGβ and NY-ESO-1. It was reported that pmel17 fused to anti-MR antibody drove cross-presentation through the MHC class I and II pathways, resulting in a CTL response [71]. A similar result was obtained from the study in which hCGβ was fused to the anti-MR antibody [72]. Finally, NY-ESO-1 fused to anti-MR antibody was also able to elicit CD4^+^ and CD8^+^ T cell responses [73]. Moreover, a comparison of non-fused with fused to anti-MR antibody NY-ESO-1 showed that non-fused NY-ESO-1 could only elicit a CD4^+^ T cell response.

Although MR expression has been found on human moDCs [74] and first conclusions and assumptions about MR expression on DCs were made based on these data, whether MR is expressed in “typical” human DC subsets is currently under debate. No MR expression has been found on human conventional DCs (cDCs), plasmacytoid DCs (pDCs), or Langerhans cells (LCs). However, there is controversy about MR expression in the dermal cDC2 subset. While there are studies that report MR expression on dermal CD1a^+^ DCs [64], other studies with the opposite results could be listed. It has been reported that more accurate analysis of so-called “dermal dendritic cells” revealed that, despite their similar morphology, they comprise at least two phenotypic populations of cells, macrophages (CD14^+^, CD68^+^, CD163^+^, CD209^+^ (DC-SIGN^+^), CD206^+^ (MR^+^)) and immature DCs (iDCs) CD1b^+^, CD1c^+^, CD11c^+^, CD14^−^, CD209^−^ (DC-SIGN^−^), CD206^−^ (MR^−^), suggesting that these are dermal macrophages, but not DCs, expressing MR [65]. A number of distinct tissue DC subsets different from “typical” DCs have been identified in inflammatory conditions. Wollenberg et al. described a subset of epidermal DCs different from LCs present within inflamed skin in atopic dermatitis and eczema [75]. These cells were named inflammatory dendritic epidermal cells (IDECs) and phenotyped as CD1a^+^, langerin^−^, CD11b^+^, CD11c^+^, with increased expression of FcεR and a lack of Birbeck granules. Moreover, MR and DC-SIGN have also been reported to be expressed on them [67]. Whereas this subset of cells was identified more than 20 years ago, data about their ontogeny and functions are limited. Another subset of putative inflammatory DCs has been isolated from the synovial fluid of arthritis patients and tumor ascites and identified as CD16^−^, CD1c^+^ DCs [76]. These cells were reported to express CD14, CD11c, CD1a, CD11b, SIRPα, FcεR1 and MR as well. Although CD16^−^, CD1c^+^ inflammatory DCs were shown to have DC-like morphology, their gene signature analysis showed that they share molecular features with cDCs and inflammatory macrophages and are closely related to in vitro moDCs. Further studies are required to clarify the ontogenetic, functional and phenotypic affiliation of the described cell subsets to DCs or macrophages. Until then, the question remains regarding whether MR is likely to be expressed on human macrophages and moDCs, or whether its expression can also be found in “typical” subsets of human DCs.

Despite the still ongoing debate about MR expression on human DC subsets, strategies of in vivo MR targeting are under development. The possibility of selective antigen delivery to MR-expressing APCs with further activation of the anticancer immune response has been demonstrated in a murine model [77,78]. In a study by He et al., transgenic mice engineered to express human MR were treated with the B11 anti-MR antibody fused to two full-size OVA molecules [77]. Further analysis revealed that immunization with B11-OVA fusion protein led to CD4^+^ and CD8^+^ T cell proliferation. However, additional simultaneous administration of TLR9 ligand CpG as an adjuvant was required to achieve CD8^+^ T cell differentiation to IFN-γ-producing CTLs. As for the humoral response, it was also reported that the simultaneous administration of the B11 antibody with B11-OVA fusion protein quantitatively and qualitatively enhanced the humoral response with its switch to Th1-type, compared to B11-OVA administration alone. This demonstrates that in vivo MR targeting with an antigen fused to an anti-MR antibody accesses the cross-presentation pathway in APCs, allowing the priming of MHC class I- and II-restricted T cell responses. Moreover, transgenic mice were also challenged with OVA-transfected melanoma cells after their immunization with B11-OVA and CpG. Thereafter, significant inhibition of tumor growth was observed in human MR-expressing mice, compared to their WT counterpart. Another similar study used the same transgenic murine model and anti-MR B11 antibody fused to another protein, hCGβ, but with the focus on the evaluation of the role of adjuvants [78]. Similarly to B11-OVA, the B11-hCGβ fusion protein was also efficiently delivered to MR-expressing APCs. Moreover, it has been demonstrated that the pretreatment of mice with GM-CSF enhances the immunity of the B11-hCGβ vaccine by an GM-CSF-induced increase in MR expression on APCs and, consequently, increased MR-targeting vaccine uptake. Administration of another adjuvant, polyinosinic: polycytidylic acid (poly I:C), which is a double-stranded RNA with TLR3 agonistic activity, in combination with B11-hCGβ, was shown to significantly increase Th1 and IFN-γ-producing CD8^+^ T cells, whereas a lack of TLR3 agonist during priming with B11-hCGβ was shown to induce even antigen-specific tolerance. Thus, pre-clinical studies proved the MR-targeting strategy mechanism and supported the idea of the clinical development of APC-targeting approaches in cancer using MR-targeting constructions.

Recently, several studies reported interesting MR-targeting approaches with a return to non-antibody targeting, but carrying antigen-encoding DNA instead of just antigens. Different reported approaches utilize different types of DNA carrier, such as polyion complexes, liposomes, and gold particles, all functionalized with mannose or mannose-mimicking ligands [79,80,81,82]. Prolonged tumor-free survival, CTL and Th1 response against B16F10 melanoma were observed in mice treated with the widely known p-CMV-MART1 DNA vaccine delivered by different types of vehicles, such as liposomes or gold particles, functionalized by a mannose-mimicking shikimoyl group [79,80,81]. Thus, non-antibody MR-targeting strategies are still relevant and under development, and further preclinical and clinical studies must be engaged to evaluate their applicability.

Mannose receptor targeting was among the first CLR-targeting strategies in cancer to be tested in clinical trials. Nevertheless, the results of only a few trials are currently available. CDX-1307 vaccine, composed of monoclonal MR-specific antibody fused to hCGβ, has been tested in a phase I clinical trial in patients with different advanced epithelial malignancies with an elevated serum level of hCGβ [44]. This vaccine was administered with adjuvants—GM-CSF pretreatment and simultaneous injections of poly I:C—as it was described above in a study with murine models (this clinical trial was performed partially by the same group which performs B11-OVA and B11-hCGβ preclinical studies [77,78]). The induction of a humoral and T cell response was observed in patients who received a combination of CDX-1307, GM-CSF and poly I:C, and no significant response was observed when CDX-1307 was administered alone (Appendix A). This study confirmed the potential of anticancer immune response induction in humans using the MR-targeting strategy and showed that it is a rational approach. A phase II trial was also initiated to test CDX-1307 along with TLR agonists and GM-CSF as a neoadjuvant therapy for patients with hCGβ-expressing muscle-invasive bladder cancer [44]. Unfortunately, this trial was terminated due to portfolio prioritization by the sponsor. Although a proof of concept has been achieved, more trials with different antigens and an assessment of clinical efficacy are required, which could encounter several problems, such as the undecided status of MR expression on DCs in humans and the appearance and description of other CLR targets on DCs that seem to be more promising in DC targeting.

#### 2.2.2. DC-SIGN Receptor Targeting

Ligands of the DC-SIGN receptor are similar to those of MR. It has also been found that DC-SIGN is often co-expressed with MR [83]. Furthermore, DC-SIGN can also recognize carbohydrates (mannose, fucose, N-acetylgalactosamine and N-acetylglucosamine) on the cell walls of bacteria, in yeasts and viruses [84]. After being captured by DC-SIGN, its ligands undergo endocytosis resulting in their involvement in the cross-presentation process. The mutation of a putative internalization motif in the cytoplasmic tail of DC-SIGN was performed as a demonstration of this function and it was found that DC-SIGN-mediated ligand-induced internalization was reduced [85]. Further, it appears that antigens internalized by DC-SIGN of moDCs are processed and presented to CD4^+^ T cells quite effectively [85]. Thus, DC-SIGN was shown to be a prospective receptor for DC targeting with anticancer vaccines.

Similar to MR, the expression of DC-SIGN on human DC subsets is somewhat controversial. It is widely utilized as a DC marker. Although its expression has indeed been found on moDCs, these cells do not represent a “typical” subset of human DCs. Being derived from peripheral blood monocyte precursors, they are likely not typical DCs but share their morphology and phenotype with DCs, macrophages and IDECs. Expression of DC-SIGN has indeed been found on macrophages and CD14^+^ cells [86]. Inflammatory dendritic epidermal cells were also recognized to express DC-SIGN [60]. At the same time, DC-SIGN expression has not been found on cDCs, pDCs, LCs, or any other “typical” subset of human DCs [60,64]. Thus, DC-SIGN should not be recognized as a pan-DC marker, but is more likely to be a macrophage marker. Nevertheless, as in the MR case, studies to determine the applicability of DC-SIGN-targeting anticancer vaccines are ongoing.

To demonstrate a ligand-based approach to target DC-SIGN, Melan-A/Mart-1 HLA-A2-restricted epitope was conjugated with Lewis oligosaccharide [87]. This conjugate was able to bind DC-SIGN with high affinity followed by internalization and cross-presentation through the MHC class I pathway [87]. Conversely, conjugation of another melanoma antigen, gp100, with glycans revealed its potential to induce a CD4^+^ T cell response through the MHC class II cross-presentation pathway [88]. Several other ligand-based approaches to DC-SIGN targeting were recently reported. Among them, gold nanoparticles functionalized with α-fucosylamide were found to induce antigen internalization as effectively as similar particles coated with Lewis oligosaccharide [89]. Such particles were found to be neutral toward maturation and IL-10 production. Another approach is to use the so-called Polyman26, which is a glycodendrimer that includes a linear rigid spacer at its core [90]. It has been shown that Polyman26 was efficiently internalized by moDCs via DC-SIGN-mediated endocytosis and then routed to early endosomal and endolysosomal compartments. This finding has implications for the development of a delivery system to target DC-SIGN-expressing APCs and facilitate antigen presentation on MHC I and MHC II. Moreover, Polyman26 showed some adjuvant properties through the upregulation of IL-1β, IL-6, IL-12 and TNFα, as well as the expression of TLR9 and CD40L.

Another effect that should be taken into account is that the antigens’ structure dramatically affects cellular routing through DC-SIGN. It has recently been reported that the internalization of antigens with different sizes and physical properties led to delivery into different cellular compartments [21]. Small soluble glycopolymers were trafficked to endosomes, while large particulate glycopolymers were localized in non-endosomal compartments where antigen processing is not performed. These findings are crucial for the design of synthetic DC-SIGN- and perhaps overall CLR-targeting vaccines.

A DC-targeting strategy with an anti-receptor antibody was also applied in the case of DC-SIGN. It has been reported that moDCs pulsed with KLH conjugated with anti-DC-SIGN antibody showed 100-fold higher activation efficacy than moDCs pulsed with KLH alone [91]. Such KLH-anti-DC-SIGN-pulsed moDCs were also able to induce the proliferation of naive T cells through MHC class I and II routes [91]. Moreover, further studies showed that targeting another region of DC-SIGN with monoclonal antibody resulted in a different mode of internalization and cross-presentation [92]. As demonstrated with MR and DC-SIGN, it is possible to regulate the way in which APCs will be targeted by the selection of the receptor ligands or antibody type. This enables the selection of which pathway of cross-presentation antigen will be involved. As such, it is possible to rationally design an anticancer vaccine depending on which response is required. However, due to similar ligands for the two above-mentioned receptors, MR and DC-SIGN, as well as due to their often co-expression [83], it is difficult to assess what contribution to immune induction each of them makes when using a particular vaccine. This aspect complicates the objective assessment of vaccine contribution, but could be overcome with DC-SIGN-specific monoclonal antibodies.

Another recent approach to DC-SIGN targeting was based on lentiviral vectors. Recombinant lentivectors were constructed by the inclusion of Sindbis virus glycoprotein engineered to be DC-SIGN-specific [93]. Received DC-SIGN-specific lentivectors were shown to specifically target mice DC-SIGN-expressing bone marrow-derived DCs and human moDCs in vitro, as well as mice DCs in vivo [93]. It was found that moDCs and mice DCs were efficiently transduced by the lentivector and induced to cross-present vector-encoding antigen with the subsequent induction of the CD8^+^ and CD4^+^ response in vitro and CTL and humoral response in mice [93]. In addition, tumor growth inhibition was observed in mice immunized with lentivector after tumor challenge [93]. Later, several modifications of the Sindbis virus glycoprotein were developed to achieve more efficient targeting and transduction of DC-SIGN-expressing APCs [94].

So far, DC-SIGN has been regarded as a possible better target for DC targeting due to its pan-DC nature in humans, a notion which mistakenly arose from human moDC and mice DC studies [95]. There is growing evidence that DC-SIGN is not expressed on “typical” human DCs at all and should be considered a macrophage marker [60]. Thus, it remains unclear which immune cell population is induced upon the application of a DC-SIGN-targeting vaccine in humans, as well as DC-SIGN’s fate as an APC-targeting vaccine target. However, clinical trials assessing DC-SIGN targeting in cancer patients have been conducted.

The above-mentioned lentiviral vectors have been tested in cancer patients with different solid NY-ESO-1-expressing tumors [45]. In this phase I clinical trial, so-called LV305 was assessed (Appendix A), which represents the so-called ZVex, a third generation lentiviral-based Sindbis virus glycoprotein functionalized vector, which is replication-incompetent and integration-deficient to reduce lentiviral-related side effects [94]. Accordingly, LV305 represents an NY-ESO-1 encoding ZVex vector. It showed efficient induction of NY-ESO-1-specific CD8^+^ and CD4^+^ T cell responses (in about half of patients) and NY-ESO-1-specific antibody response, but only in 8% of patients, with measurable responses occurring as long as two years after treatment. However, only two patients achieved at least a partial response, while 20 patients (51.3%) had stable disease. Overall, while well-tolerated, LV305 showed moderate efficacy and was suggested to be applied in combination with ICI, since it was noted that it took several months for the anti-NY-ESO-1 CD8^+^ T cell response to occur [45].

Another phase I clinical trial assessing antibody-based DC-SIGN targeting in cancer patients has been completed with available results (Appendix A) [46]. In this clinical trial, patients with stage IV melanoma received so-called Lipovaxin-MM polycomponent vaccine consisting of MM200 melanoma cell line derived plasma membrane vesicles (PMVs) containing melanoma antigens (gp100, tyrosinase and MART-1). The PMVs were modified with POPC/Ni-3NTA-DTDA liposomes, IFNγ and DMS5000 antibody. DMS5000 is a VH-domain antibody, which is specific to the DC-SIGN moiety. DMS5000 efficacy as a vaccine targeting method has been demonstrated in preclinical studies in melanoma-bearing mice [96]. The DMS5000-containing vaccine showed the effective induction of anti-tumor T cells in mice [96]. However, Lipovaxin-MM failed to show any significant cellular or humoral immune response in the patients of the described clinical trial, although it was safe and some clinical response barely associated with the immune response was observed [46]. However, it should be noted that the Lipovaxin platform can easily be adapted for DC targeting to other DC receptors with receptor-specific antibodies.

#### 2.2.3. DEC205 Receptor Targeting

DEC205 is another CLR that recycles through late endosomal or lysosomal compartments, resulting in its involvement in antigen cross-presentation and suitability for in vivo antigen-targeting [97]. In contrast to MR and DC-SIGN, DEC205 expression was found on different types of human DCs, including “typical” DCs [41,60,98]. In addition, DEC205 expression was found at intermediate levels on B cells and low levels on NK and T cells [98]. The major function of this receptor has been suggested to be binding apoptotic and necrotic cells for the further uptake and cross-presentation of debris-associated antigens by DCs [99].

The only approach that has been described to target DEC205 is antigen fusions with anti-DEC205 antibodies, but not with its ligands. The possibility of achieving an anti-tumor immune response by targeting DEC205 with receptor-specific antibodies has been shown in numerous ex vivo studies with human and mice cells and in vivo studies in a murine model. DEC205 targeting in mice was shown using anti-DEC205 antibodies or single-chain variable fragments (scFv) fused to different antigens or using fusion-encoding DNA vaccines [100,101,102]. Besides the high specificity to DCs, such vaccines have been shown to effectively induce anti-tumor CD8^+^ and CD4^+^ T cell and humoral immune responses in mice in a lower dose, compared with the administration of non-targeted antigens [103]. However, to achieve an immune response, but not immune tolerance, an additional application of adjuvants was required, which is not a surprise. Particularly, experiments with chimeric anti-DEC205 antibody fused to myelin oligodendrocyte glycoprotein showed potential therapeutic benefits, as its administration blocked autoimmune responses and disease symptoms of experimental autoimmune encephalomyelitis, a murine model of multiple sclerosis [104]. It has been reported that DEC205 targeting of antigens without the application of any adjuvant can be utilized even for the induction of antigen-specific immune tolerance to ameliorate disease severity in models of autoimmune diseases, such as diabetes, inflammatory bowel disease and arthritis [105]. The administration of DEC205-targeted antigens with CpG, poly I:C, or CD40L as adjuvants lead to the reverse situation—the induction of antigen-specific CD8^+^ and CD4^+^ T cell responses, including pro-inflammatory cytokine production, thus showing the obligatory application of adjuvants in the context of anticancer vaccines [106,107]. Other more effective adjuvants, e.g., c-di-CMP, have been tested and reported to be superior at inducing humoral and cellular immunity compared with traditionally utilized poly I:C and CpG [108]. Moreover, this targeting strategy could also be applied to adjuvants. It has been shown that anti-DEC205 antibodies coupled with both antigen and CpG or poly dA:dT improve the CD4^+^ and CD8^+^ T cell responses compared with untargeted adjuvant administration [109,110]. Such an adjuvant-targeting strategy might reduce side effects from the activation of other cells, since DEC205 was found to be expressed not only by DCs [98,111].

To translate DEC205 targeting into humans, a transgenic murine model with human DEC205 was used to show DEC205 targeting with human anti-DEC205 antibodies [112]. In this work, HIV Gag p24 fused with anti-DEC205 antibody was successfully delivered and cross-presented to T cells [112]. As for clinical trials, DEC205 targeting was assessed using CDX-1401 agent, which is a full-length NY-ESO-1 tumor antigen fused with human anti-DEC205 monoclonal antibody [47]. In this phase I clinical trial, 45 patients with different advanced NY-ESO-1-expressing malignancies received CDX-1401 with subcutaneously administered adjuvant, either poly I:C, resiquimod, or both (23 patients). The induction of humoral, CD8^+^ and CD4^+^ NY-ESO-1-specific T cell responses has been observed with no signs of toxicity. Thirteen patients achieved stable disease, including two patients with tumor reduction. What is of interest in this trial is that six patients with melanoma received anti-CTLA4 administrations within three months of the last vaccine treatment. Four of them were reported to reach a partial response or complete response, which is greater than the expected 15% response rate for ipilimumab monotherapy, which indicates a vaccine and ICI synergetic effect [113]. This observation supports the rationale of the approach mentioned at the beginning, i.e., combination cancer therapy with a vaccine and ICI is superior compared to vaccine or ICI monotherapy. Such a combination therapy might be the method of choice in the future non-invasive treatment of cancer.

DEC205 is likely to be the most prospective candidate for DC targeting among the three already discussed well-studied CLRs, including MR and DC-SIGN. Antigens targeted to DEC205 are engulfed and involved in their recycling in an endocytic manner. Moreover, antigen targeting to DEC205 leads to their engulfment by DCs specifically, but not macrophages or other APCs, unlike in the MR- or DC-SIGN-targeting case. A large body of evidence illustrates the effectiveness of DEC205 targeting in mice to prevent the development of infectious diseases or cancer. The results of a few clinical trials look very promising. Accordingly, the results of all these studies combined promote the recognition of DEC205 as the most deserved target of the in vivo DC-targeting strategy.

#### 2.2.4. Other C-Type Lectin Receptor Targeting

The most likely current candidates among CLRs to enter further clinical trials assessing in vivo targeting of DCs in cancer, despite ongoing debates about their DC-specific expression, are the receptors described above. Next to these CLRs with well-defined carbohydrate specificity, DCs are known to express other CLRs that have poorly defined ligands and functions. These CLRs have also been studied for DC-targeting purposes and several of them, like Clec9A and Clec12A, could be listed as examples.

Gene expression profiling of murine DC subtypes led to the identification of Clec9A and the human ortholog, known as well as DNGR1, was identified based on the similarity of their sequences [114]. This has been described to be specifically expressed by murine CD8α^+^ DCs. In humans, BDCA3^+^ DCs have been described as equivalents of murine CD8α^+^ DCs and Clec9A is expressed specifically by this subtype of DCs [115,116]. A few other studies also reported its presence on a CD14^+^ CD16^−^ subset of monocytes [117].

The CD8α^+^ DCs have a superior ability to cross-present exogenous antigens on MHC class I and therefore activate CD8^+^ T cells [118]. Moreover, CD8α^+^ DCs are the major producers of IL-12 when activated and appear to be able to induce an inflammatory Th1 response [119,120]. Later, it was reported that CD8α^+^ DCs are also effective activators of CD4^+^ T cells and this is presumably the role they play in enhancing the humoral response [114]. Animal studies showed that antigen targeting to Clec9A induces strong anti-tumor immunity via the activation of CD8^+^ and CD4^+^ T cells [121,122]. Additionally, Clec9A targeting in mice stimulated a strong humoral response even without adjuvants [123,124]. However, the selection of antibodies that target Clec9A is crucial in this context. Considering the functional features of murine CD8α^+^ DCs and the characterization of Clec9A^+^ BDCA3^+^ DCs that resemble murine CD8α^+^ DCs in phenotype and function [115], Clec9A targeting is currently one of the most promising in vivo DC-targeting strategies, together with DEC205 targeting. Indeed, several studies report Clec9A^+^ BDCA3^+^ DCs’ ability to cross-present antigens more efficiently than that of other human DC subsets [125,126].

In vitro studies with human cells showed the significant effectiveness of antigen priming to CD8^+^ and CD4^+^ T cells after Clec9A targeting with monoclonal anti-Clec9A antibodies [61]. Compared to DEC205, only BDCA3^+^ DCs have been found to express Clec9A selectively, while DEC205 expression has been found on moDCs, BDCA1^+^ (CD1c^+^), CD16^+^ and BDCA3^+^ (CD141^+^) APCs as well [61]. However, opposite to murine Clec9A targeting, the application of TLR ligands as adjuvants was required for proper DC maturation and activation of pro-inflammatory cytokine production in the case of human cells [61,123]. This finding confirms the above expressed necessity of adjuvant application for DC targeting, or the utilization of molecules with both adjuvant and DC-targeting properties simultaneously. The results of in vitro studies are in line with the results of Clec9A and DEC205 targeting in humanized mice. It has been demonstrated that the Clec9A-targeted antigen was efficiently delivered to the cross-presentation pathway in BDCA3^+^ DCs in vivo, whereas DEC205-targeted antigen was delivered to both BDCA3+ and BDCA1^+^ (CD1c^+^) DCs [127]. Moreover, Clec9A-induced cross-presentation to CD8^+^ and CD4^+^ T cells was more efficient than DEC205-induced. The authors explained such low-level cross-presentation efficacy in DEC205 targeting by the broad expression of DEC205 and the ability of anti-DEC205 antibodies to target a larger number of DCs, including CD1c^+^ DCs and pDCs [127]. However, there may be some instances in which the broader specificity of DEC205 could be advantageous.

As there are still unknown ligands for Clec9A, its targeting in almost all studies was performed using Clec9A-specific antibodies. Recently, an analysis of the binding moieties of Clec9A was performed and a peptide that interacts with Clec9A was identified [128]. Accordingly, so-called WH synthetic peptide was produced and tested for Clec9A specificity and the ability to enhance the antigen-specific immune response. It was found that WH peptide was indeed able to bind Clec9A specifically in experiments. Moreover, a conjugate of WH peptide to the immunodominant OVA epitope administered simultaneously with murine CpG ODN as an adjuvant significantly enhanced the activation of OVA-specific CD8^+^ T cells by Clec9A^+^ DCs and decreased metastasis formation in a murine B16-OVA melanoma lung-metastasis model. Another recent study reported a different approach to Clec9A targeting using nanoparticle delivery. It is known that Clec9A binds F-actin from necrotic cells, allowing the cross-presentation of dead cells by Clec9A^+^ BDCA3^+^ DCs [129,130]. In another study, Zeng et al. [131] synthesized Clec9A-specific oil-in-water nanoemulsions with either encapsulated full-size OVA antigen, OVA_257–254_ CD8 and OVA_323–339_ CD4 epitopes, or a mixture of six neoepitopes derived from somatic melanoma mutations. They achieved Clec9A-specificity by the functionalization of nanoemulsion with anti-Clec9A antibodies, polymerized F-actin, or WH peptide. It has been demonstrated that anti-Clec9A nanoemulsion bearing OVA or its CD8 and CD4 epitopes can efficiently induce CTL and T helper responses, as well as inhibit tumor growth in mice bearing PyMT-ChOVA breast cancer cells. Moreover, no additional adjuvants were required, since anti-Clec9A nanoemulsion was observed to be self-adjuvant. Similar results were obtained when F-actin-nanoemulsion or WH-nanoemulsion, both bearing full OVA or neoepitopes, were applied. Overall, Clec9A represents another CLR that is worth using in DC targeting, along with DEC205. Thus, the translation of the Clec9A-targeting strategy into human studies and clinical trials is expected in the near future.

It is worth mentioning other CLRs—Clec4A/DCIR and BDCA2—which have roles in mediating T cell responses. Mice have been found to express multiple DCIR molecules; however, humans express only one type [132]. Dendritic cell immunoreceptor has a broad carbohydrate specificity for mannose and fucose [133]. In humans, DCIR is expressed on B cells and different DC subsets, including pDCs. Targeting DCIR on human cDCs and pDCs resulted in antigen presentation in vitro [134,135]. However, Clec4A is the group of CLRs with an ITIM (immunoreceptor tyrosine-based inhibitory) motif in their cytoplasmic tail [136]. Consequently, DCIR targeting simultaneously suppressed TLR9-mediated production of type I IFNs by pDCs and TLR8-mediated production of pro-inflammatory cytokines by cDCs [135,137]. Therefore, DCIR is associated with homeostasis and inflammation control and is therefore not very well suited to DC targeting in the anticancer vaccination context. Continuing with pDC targeting, BDCA2, which is specifically expressed by pDCs, has been utilized [138]. Although this resulted in antigen cross-presentation, similar to DCIR, BDCA2 targeting simultaneously inhibited the production of type I IFNs by pDCs [138]. Considering all of these functions, DCIR and BDCA2 targeting strategies are superior in the promotion of immune tolerance, which may be useful in treating autoimmune diseases.

Another ITIM-containing CLR Clec12A has been proposed to target DCs. It has been demonstrated that Clec12A is expressed by human BDCA1^+^ DCs, BDCA3^+^ DCs, pDCs, and moDCs, but can also be found on B cells, CD8+ T cells and NK cells [139,140,141]. Although containing an ITIM inhibitory motif, Clec12A targeting in human cells has been shown to drive effective cross-presentation and activation of specific CD8^+^ and CD4^+^ T cells [139]. In addition, Clec12A targeting has been shown to enhance antibody response in the presence of adjuvants [142]. However, Clec12A is not considered a superior DC-targeting molecule since it was found to be less efficient in CTL response induction compared to Clec9A and DEC205 [122].

In summary, CLR targeting strategies performed well in in vitro and in vivo studies and proof of the CLR targeting mechanism has been well documented. However, little can be inferred from a few phase I clinical trials. More clinical trials are expected in this field and more time is needed to obtain a strong proof of concept of the CLR targeting strategy in the anticancer vaccination field. For now, it makes sense to focus more on DEC205 and Clec9A targeting among all other CLRs in the case of CLR targeting strategies.

### 2.3. Scavenger Receptor Targeting

Scavenger receptors were initially described as receptors for modified or oxidized forms of lipoproteins [143]. Later, the range of ligands for SRs was expanded to other molecules, such as LPS, fucoidan, polyanionic ligands, advanced glycation end products, nucleic acids and heat shock proteins (HSPs) [40,143]. The SR family consists of eight classes (A–H), most members of which are expressed on the surface of APCs, including DCs. Besides scavenging functions, SRs enable antigen endocytosis and their involvement in cross-presentation through the MHC class I and II pathways [144,145]. Furthermore, evidence has demonstrated that TLR signaling is finely tuned by SRs expressed on APCs. Many SR ligands are also recognized by and trigger cell signaling through TLRs. It has been demonstrated that SRs internalize such ligands to render them accessible to TLR3, TLR7/8 and TLR9 localized in the cytosol [146,147]. Taking into account such findings, SRs were also proposed as a potential target for anticancer vaccine delivery to DCs. Most SR targeting strategies do not differ from those previously described in relation to other DC receptors. The recipe is to conjugate, fuse or construct a complex of the SR ligand or anti-SR antibody with a well-chosen TSA. Protein maleylation is used to achieve the ability of the protein to bind SRs. It has been demonstrated that maleylated OVA was able to bind SRs, enhancing its cross-presentation and CTL priming [144]. Moreover, maleylated antigens were shown to be more immunogenic than non-maleylated [148]. Utilizing anti-SR antibodies was also shown to be useful in SR targeting. It has been demonstrated that influenza viral nucleoprotein (Flu.NP) fused to either anti-LOX-1 or anti-Dectin-1 antibodies was able to promote a Flu.NP-specific CD8^+^ and CD4^+^ T cell response on in vitro human peripheral blood mononuclear cells (PBMCs), with superior activation of CD4^+^ T cells [149]. A nanoparticle approach was also applied to target SRs on DCs. High-density lipoprotein (HDL)-mimicking nanostructures have been developed to target B1 class SRs (SR-B1) [150]. Functionalization of HDL-mimicking particles with fusion peptide, consisting of OVA_323–339_, OVA_257–264_ and hgp100_25–33_, gave them the ability to deliver this peptide to lymph node DCs through SR-B1 internalization in mice [151]. Peptide-functionalized HDL-mimicking nanoparticles showed high efficacy in cytotoxic CD8^+^ T cell activation and the prevention of tumor growth in mice, even without adjuvants. However, further functionalization of nanoparticles with CpG dramatically enhanced anti-tumor efficacy.

An interesting SR targeting strategy utilizing HSPs as targeting molecules has been proposed. Heat shock proteins are ubiquitous and perform protein-maintaining functions intracellularly with their highly efficient protein- and peptide-binding ability [152]. Besides intracellular functions, HSPs also possess extracellular functions which include immunomodulatory functions and protein trafficking to APCs [153]. It has been demonstrated that cells under stress or suffering from necrosis appear to be the source of HSP-host protein complexes [154,155]. Such HSP-host protein complexes discharged into the extracellular environment could be captured and internalized by DCs, followed by their processing and cross-presentation [156]. The uptake of HSP-host protein complexes is mediated by SRs, specifically by LOX-1, FEEL-1 and SREC-1 [40,157]. Thus, HSPs are natural vehicles for antigens that are expressed by cells which undergo necrosis or stress, including cancer cells. Therefore, a number of strategies utilizing HSPs to target the SRs of DCs have been developed [158]. Different studies reported different variants of HSP-based vaccines designed to target SRs, which included the harvesting of autologous HSP-TAA complexes [159], in vitro constructed HSP-TSA complexes [160,161] and fusions of HSPs to cancer peptides [162]. Eventually, all of these methods aim to develop a DC-targeting vaccine that will activate DCs in vivo after its administration to the patient. To the best of our knowledge, only HSP-containing approaches were tested in clinical trials of anticancer vaccination among SR-targeting strategies. Numerous clinical studies have shown that vaccines containing HSPs as a targeting molecule for SRs can effectively induce T cell response and increase overall and progression-free survival. Several of them are reviewed in Appendix A.

### 2.4. Targeting Dendritic Cells Using Genetically Modified Lymphocytes

It is known that defined subsets of T cells can migrate to secondary lymphoid organs (SLO) after being exposed to a short TCR stimulation by a lack of IL-12 and IL-4 [163]. Such lymphocytes, being primed, do not acquire an effector function and remain in a non-polarized state. After reaching SLOs, non-polarized lymphocytes begin to proliferate and may acquire an effector function more promptly and vigorously, representing memory cells for an immediate secondary response [163,164]. Further, it has been noted that the adoptive T cell transfer of genetically modified lymphocytes (GML) triggers a specific immune response in a vaccine-like mode against a transgene transfected to GMLs [165]. Accordingly, whether GMLs transduced with TAA are indeed able to induce a TAA-specific immune response became a matter of interest. Ovalbumin-expressing GMLs were tested in a murine model and it was found that they can induce a CD8^+^ and CD4^+^ T cell response against OVA and DCs which were responsible for T cell priming [166]. Thus, the observed phenomenon was translated into human studies to test SLO-resident DC targeting with GMLs expressing TAAs. Genetically modified lymphocytes transduced with MAGE-A3 were applied in melanoma patients [55,167]. It has been shown that MAGE-A2-transduced GMLs can induce MAGE-A2-specific T cell effectors [167]. Additionally, a significant association with increased overall survival in treatment-responsive patients has been observed. However, only a few patients responded to therapy (six out of 22 patients) [55].

Secondary lymphoid organ DC targeting by GMLs therefore represents a feasible approach to DC targeting for cancer treatment. However, there are major disadvantages compared with other in vivo targeting approaches. Although it is considered an in vivo targeting strategy, the preparative procedure requires invasive manipulations to harvest autologous lymphocytes. Further steps of genetic modifications and cell maintenance also intensify the preparation of such vaccines for individual patients.

### 2.5. Receptor-Mediated Maturation, Activation, and Migration of Dendritic Cells (Adjuvants)

The involvement of internalized antigens in processing and cross-presentation via the MHC class I and II pathways is the central mechanism utilized in DC-targeting vaccination. However, DC receptors also contribute towards the maturation and activation of DCs. Prior to encountering antigens, DCs represent an immature phenotype characterized by a high expression of receptors for antigen uptake and a low expression of co-stimulatory molecules and chemokine receptors, and can be described as biologically equipped for antigen capture and internalization [168]. However, antigen uptake alone is not enough for DCs to start the maturation process. To initiate maturation, DCs should also receive some signals through their environmental sensors, so-called PRRs. Such sensors recognize PAMPs and DAMPs as signals to start maturation. Moreover, it has been demonstrated that exposure of iDCs to maturation signals leads to an initial upregulation of antigen uptake, followed by a rapid termination of the antigen capturing function and transition to the mature phenotype [169,170,171]. Such a process includes the upregulation of chemokine receptors, surface MHC I and MHC II molecules, co-stimulatory molecules and the secretion of cytokines to regulate the type of T cell response [168]. Conversely, the uptake of antigens in a “sterile” condition without the presence of PAMPs or DAMPs but in the presence of anti-inflammatory cytokines may lead to the development of an immune-tolerogenic DC phenotype [24,172]. It is now well known that DCs also mediate T cell homeostasis and peripheral tolerance [26,173,174]. Thus, for a rational DC-targeting vaccine composition, the certain dependence of DCs on maturation signals must be taken into account. The function of such signals can rely on adjuvants that could bind DC environmental sensors and enhance their antigen uptake and maturation abilities. Interestingly, several DC-targeting molecules represent PAMPs or DAMPs at the same time. This allows the use of single-component vaccines with one molecule serving as the ligand for the DC antigen-uptake receptor and for the activation of DC maturation and activation at the same time.

The most described PRRs and thus the most utilized for artificial DC maturation and activation are TLRs. One of the major players in TLR signaling is myeloid differentiation primary response 88 (MyD88) protein, except for TLR3 and TLR4 signaling, in which Toll/IL-1 receptor domain-containing adaptor protein inducing interferon-β (TRIF) is essential [175]. Whereas TRIF is essential for the TLR4-mediated production of inflammatory cytokines, activation of MyD88 stimulates the MAPK and NFκB pathways, which are required for the induction of pro-inflammatory cytokine production, surface co-stimulatory molecule expression and lymph node homing CC chemokine receptor 7 (CCR7) expression in DCs [176]. As such, pattern recognition through MyD88 signaling is acknowledged to be crucial for the induction of appropriate immune responses via the release of pro-inflammatory cytokines, regulation of the innate response and polarization of T cells and humoral immunity. Different substances that represent TLR agonists have been used to target TLRs in vivo in anticancer vaccine compositions. The most applied are resiquimod and different derivatives of poly I:C, TLR7/8 and TLR3 agonists, respectively, which have been used in different clinical trials accompanied by different types of in vivo APC-targeting vaccines [44,47]. It was reported in another murine model study that the application of TLR9 agonist CpG, which is short single-stranded DNA, as a APC-targeting vaccine adjuvant was necessary to achieve CD8^+^ T cell differentiation to IFN-γ-producing CTLs [77]. In addition to the widely utilized resiquimod, poly I:C and CpG, some other alternative TLRs ligands, such as poly dA:dT or c-di-CMP, have also been applied and were shown to be superior in some cases [106,107,108,110]. The application of TLR adjuvants has been shown to even be obligatory, since utilizing APC-targeting vaccine administration without any adjuvants promoted antigen-specific immune tolerance [44,78]. It was proposed that this phenomenon could be utilized in the case of autoimmune diseases to ameliorate their severity [104,105].

Another type of adjuvant that can be utilized is CD40 receptor ligands. CD40 is a TNF receptor superfamily member expressed on APCs, including DCs, B cells and monocytes, as well as by many other non-immune and tumor cells [177]. In the context of DCs, CD40 activation is required for the enhancement of surface co-stimulatory and MHC molecule production, pro-inflammatory cytokine production and enhanced T cell triggering. Physiologically, CD40 signaling in DCs is provided by CD40L expressed on the surface of activated T helper cells. Thus, utilizing CD40 ligands as a DC-targeting vaccine adjuvant has also been proposed. Initially, recombinant CD40L itself was tested in cancer patients with promising results [178]. Further, CD40 agonist monoclonal antibodies have been developed and applied as immunotherapy in cancer clinical trials [179,180]. Regarding the described functions of CD40, anti-CD40 antibodies have also been utilized in compositions of DC-targeting vaccines to enhance their efficacy [106,107].

Similarly to antigens, adjuvants can also be delivered specifically to a certain receptor or APC subtype. Initially, co-delivery of OVA and CpG both conjugated to the same DEC205-specific antibody to DCs has been shown to be superior in CTL induction in comparison to anti-DEC205-OVA conjugate mixed with soluble CpG [109]. Applying such an adjuvant targeting approach might reduce adjuvant-related side effects since they are non-specific ligands for different types of cell. However, a decrease in the receptor-targeting specificity of vaccines has been reported due to the presence of a nucleic acid moiety (CpG), which allows vaccine uptake by other than the targeted receptors [109]. Thus, there is a question of balance, if the activation of innate immunity properties (to induce activation and maturation of DCs), or adaptive immunity (to enhance DC-specific antigen processing and T cell priming) are required. To overcome this issue, the simple division of such a single-component vaccine into two simultaneously administered components—receptor-specific adjuvant-antibody conjugate and receptor-specific antigen-antibody conjugate—can be performed [110].

The utilization of cytokines has also been proposed to improve in vivo DC-targeting vaccination. As an example, GM-CSF pretreatment can be listed. It has been shown that GM-CSF pretreatment led to an increase in the numbers of circulating myeloid DCs, proliferative CD4^+^ and CD8^+^ T cells, recruitment of CD8^+^ T cells to the tumor and a decrease in the numbers of myeloid-derived suppressor cells (MDSCs) [181,182]. Moreover, it has been shown in mice that GM-CSF pretreatment can enhance vaccine efficacy by increasing antigen uptake receptor expression, such as MR [78]. Thus, although GM-CSF pretreatment does not affect the DC maturation process or T cell cytotoxicity, it increases the chances a vaccine will be delivered and primed via the recruitment of DCs and T cells.

Elaborating on the cytokine-based vaccine enhancement strategy, applying CD40L- and GM-CSF-transfected cancer cell lines DC activators and recruiters has been suggested. For this purpose, K562 chronic myelogenous leukemia cell line has been stably transfected and engineered to secrete GM-CSF and express CD40L on its surface [183]. Such a bystander cell line has been observed to significantly enhance anti-tumor T cell responses in vitro if mixed with human autologous tumor cells, compared to autologous tumor cells alone [183]. The consistency of the approach was confirmed in a study with murine models and the use of a B78H1 bystander cell line transfected with GM-CSF and CD40L [184]. Consequently, a number of clinical trials have been conducted to test the so-called GM.CD40L bystander cell line, which is a K562 MHC-negative GM-CSF-secreting CD40L-expressing cell line, in the formulation of tumor cell-based vaccines in lung adenocarcinoma patients [185,186,187]. It was shown that GM.CD40L bystander cells are safe, recruit and activate DCs, and enhance tumor-specific T cell responses. However, no significant associations between vaccine immunogenicity and clinical outcomes were found.

Last but not least, an approach that is worth mentioning is the utilization of bifunctional substances that possess targeting and adjuvant properties simultaneously. The application of such substances may remove the requirement of an adjuvant in vaccine composition, thereby enabling the use of single-component vaccines. Scavenger receptor ligands may perform such functions, since some SRs represent PRRs as well [188]. Moreover, SRs have been shown to cooperate with TLR-mediated signaling and cytokine production [146,188,189]. Heat shock proteins are one example of SR ligands that can be utilized to achieve DC targeting and activation at the same time [157]. Different nanosubstance-based approaches might be also helpful, since several nanosubstances utilized to deliver tumor antigens to a specific DC receptor, such as glycodendrimers and nanoemulsions, have been shown to be self-adjuvant [90,131,190].

In vivo DC targeting has huge potential in the anticancer vaccines field. Alongside high immunogenicity and safety, vaccines that target DCs in vivo are constructed in such way that they only need to be rationally administered. Being in a DC-rich region of the body, such vaccines will further be internalized by natural or conventional subsets of DCs. Accordingly, all of the processes of DC activation, maturation, migration and CTL priming will occur naturally, in the right microenvironment. This limits all invasive manipulation of the patient to a minimum. Moreover, in vivo DC-targeting vaccines are less complicated, less labor-intensive, less time-consuming and less expensive to prepare among DC-based vaccines. In vivo DC targeting is a relatively recent approach and there have not been enough clinical trials completed to enable a robust comparison regarding ex vivo DC loading. However, a start has been made and several studies indicating the great potential of in vivo DC targeting in anticancer vaccination have been completed. Many new trials are already engaged and expected in the very near future.

## 3. Ex Vivo Dendritic Cell Loading

With the development and improvement of technologies and methods, it became possible to cultivate DCs ex vivo, which subsequently led to a breakthrough in the understanding of DCs. Numerous DC subtypes, their receptors, pathways and functions have been described, helping to understand the nature of DCs and their role in innate and adaptive immunity. With such knowledge and methodology, ex vivo DC activation in the field of DC-based vaccination was proposed and tested in clinical trials (Appendix A). Ex vivo strategies exploit the same paradigm of the possibility for DCs to cross-present exogenous antigens on MHC class I and II molecules for different T cell subtypes. However, these strategies are based on a completely different approach to antigen delivery to DCs. This approach involves direct ex vivo loading of antigens into autologous-derived DCs with an efficient DC stimulation through a “maturation cocktail”, which typically consists of a combination of pro-inflammatory cytokines and Toll-like receptor agonists (Figure 2) [191]. Besides targeting DC receptors, the ex vivo approach provides the possibility of applying a wide spectrum of more efficient antigen loading methods that cannot be applied in vivo. Ex vivo strategies of antigen loading to DCs include direct loading of TSAs or their peptides [192]. Even whole tumor lysates could be loaded into DCs [193]. Moreover, the transduction of DCs with viral vectors and mRNA, which encodes TSAs, could be applied [194,195]. Recently, another interesting approach has been proposed. It has been demonstrated that by analogy with hybridomas, it is possible to construct chimeric cells represented by the fusion of DCs with tumor cells [196]. Such DC/tumor fusion cells (FCs) may cross-present a full palette of TSAs. This approach provides an opportunity to bypass the restrictions arising from the difficulty of TSA identification and the identification of epitopes corresponding to a patient’s MHC allele variant.

### 3.1. Methods of Ex Vivo Dendritic Cell Generation

The major difference of in vivo DC targeting compared to ex vivo DC loading is that antigen uptake by DCs and the DC maturation process take place in a natural microenvironment, or in an artificially created environment, respectively. Therefore, in the case of ex vivo strategies, autologous DCs first need to be harvested for in vitro cultivation. Despite the fact that there are many DC subsets, they are quite a small population of leukocytes.

To date, there are two dominant methods of DC-like cell or DC generation that require different sources of progenitors—monocytes or CD34^+^ hematopoietic precursors. The differentiation of monocytes from autologous PBMCs to so-called moDCs is the most common method. However, moDCs do not represent cDCs, but are more likely to share a phenotype with DCs and macrophages. To generate moDCs, monocytes could be routinely obtained from peripheral blood using the leukapheresis procedure followed by the washing of monocytes, or by other magnetic bead-based purification methods of selection [197,198]. Obtained monocytes are differentiated into immature moDCs during five to seven days of cultivation with GM-CSF and IL-4. Another method of iDC generation is to use CD34^+^ hematopoietic precursors. The sources of CD34^+^ precursors for this method could be cord blood, bone marrow aspirate, or the patient’s peripheral blood. However, patients should receive GM-CSF administrations prior to leukapheresis procedures for peripheral blood stem cell mobilization. The obtained CD34^+^ hematopoietic precursors are differentiated into iDCs during seven to 12 days of cultivation with a combination of different cytokines, depending on the specific method: GM-CSF, SCF, Flt3L, TPO, TNF-α, IL-3, IL-4, IL-6 and the antagonist of the aryl hydrocarbon receptor (StemRegenin 1) [199,200,201]. Consequently, a resulting mixture of moDCs, DCs that are phenotypically similar to LCs, XCR1^+^ DCs, pDCs and a large proportion of myeloid cells at different stages of differentiation can be obtained [199,201,202]. Comparing the two approaches of moDCs and CD34^+^-derived DCs, the former is more intensively used due to its easier derivation procedure. Moreover, moDCs are more homogeneous and completely differentiated compared to the CD34^+^-derived mixture of DCs. However, moDCs are the least similar to cDCs phenotypically [203]. Preclinical and clinical studies revealed that using DCs phenotypically close to cDCs or their combination with pDCs may result in a stronger clinical response [204,205]. It has been demonstrated that CD34^+^-derived DCs and their LC portion in particular were more effective in stimulating a CTL response in vitro than the moDC portion of the same CD34^+^-derived DC pool [206].

The usage of the above described methods of iDC generation is a time-consuming process. Furthermore, moDCs do not correlate well with in vivo cross-presenting cDCs. Direct isolation of cDCs from PBMCs could be a potent alternative for moDCs and CD34^+^-derived DCs. A major advantage of cDC isolation is the absence of the requirement to undergo a time-consuming and exhausting DC differentiation process. Instead, cDCs require only brief ex vivo exposure for activation and can then be loaded with antigen [207]. Usually, the whole procedure takes less than 24 h [207]. An obstacle to the use of isolated cDCs is their low yield due to their relatively scarce population in peripheral blood. The method allows 1 × 10^7^–1 × 10^8^ DCs to be harvested in a single apheresis [207]. Nonetheless, following the isolation of PMBCs, cDCs of the preferred subset may be enriched using commercially available kits or by cell sorting [208]. Moreover, it has been demonstrated that cDC vaccines are safe and capable of inducing a de novo immune response at a number of DCs as low as 3–10 × 10^6^ [207].

Another approach to generate DCs ex vivo and to overcome the limitations of phenotypically controversial moDCs and the paucity of cDCs would be the use of DC-like immortalized cell lines. Several human DC models were developed and described for their potential application as therapeutic vaccines [209]. The overwhelming majority of them are leukemia cell lines that can be infinitely maintained and differentiated into DC-like cells in vitro when required [209,210]. For instance, among the human leukemia cell lines suitable for differentiation into DC-like cells, MUTZ-3 cells were established as the most promising and come close to mimicking a DC-like phenotype. In particular, for antigen presentation and T cell stimulation, the MUTZ-3 cell line is a model of choice among cell lines [211]. They can also be stably maintained at the pre-DC stage in culture and provide an adequate amount of inducible DC-like cells [211,212]. One may raise a question about the rationale and safety of using allogeneic DC-like cells as vaccines, as they may fail to induce a proper effect or elicit major adverse events. Complete or at least partial human leukocyte antigen (HLA) matching is required for the application of allogeneic DC vaccines, which would be a limiting factor. However, it has been shown that allogeneic DCs isolated from cord blood can be as potent in immunogenic terms as autologous moDCs [213]. It has recently been reported that an DCP-001 allogeneic vaccine, which consists of the DCOne acute myeloid leukemia cell line differentiated into mature DC-like cells, was tested in a phase I clinical study and found to be safe and able to generate a cytotoxic and humoral response [214]. Accordingly, it can be claimed that the use of allogeneic DC vaccines is feasible, but further testing is necessary.

Finally, the establishment of a stable engineered immortalized DC line, designated as ihv-DC, has been reported recently [215]. In this study, PBMCs were transduced with a lentiviral construct encoded TAX protein from human T cell leukemia virus type 2 (HTLV-2). The TAX protein from HTLV-2 is known to activate various cellular signaling molecules, including NF-κB, TRAF6, Stat3 and PI3-kinase. These molecules are crucial to DC maturation and activation [216,217]. Transduced PBMCs were then cultured continuously for three weeks in the presence of IL-2 followed by the selection of CD3^−^ cells to deplete T cells. Eventually, TAX^+^ CD3^−^ cell cultures from transduced PBMCs were continuously grown over six months in the presence of IL-2 and then their phenotype was tested. Expression of CD11c, CD141/BDCA3 and CD205, as well as the DC maturation and activation molecules CD83, CD80, CD86, CD70, CCR7 and HLA-DR, was detected on the obtained cells. Therefore, ihv-DC cells represented the CD11c^+^ CD141/BDCA3^+^ CD205^+^ phenotype of activated and mature DCs. To analyze the stimulatory ability of ihv-DC cells to induce a CTL response, they were transfected with different TSAs and then exposed to naive CD3^+^ PBMCs. Further experiments revealed that naive CD3^+^ PBMCs were able to generate antigen-specific CTLs that induce cytolysis of target cells. The authors highlighted several features of the engineered ihv-DC cells, including that they were constitutively activated and mature without the need for additional stimulation, they constitutively expressed abundant amounts of co-stimulatory receptors for CD8^+^ T cell proliferation and they were immortalized, although their immortalization efficiency was about 20% [215]. In summary, although the described approach of engineered immortalized ihv-DC cell line obtainment is intriguing, many questions arise. One of them is that it seems impossible to load ihv-DC cells with antigen by any other manner except by genetic modification to force them to produce it. This is due to the fact that ihv-DC cells are already mature and thus may no longer regulate antigen uptake under this phenotype. This situation may restrict the use of this model for DC vaccine production. Whether or not it is possible to produce precisely a DC but not DC-like immortalized cell line is intriguing and requires further investigation and clinical trials.

### 3.2. Dendritic Cell Loading with Peptides, Proteins and Tumor Cell Lysates

After the generation of iDCs, they are ready for effective antigen loading, followed by an artificial maturation procedure. The most commonly utilized methods of ex vivo DC loading are pulsing with peptides corresponding to the patient’s MHC class I and II molecules, TAAs, or whole tumor cell lysates (Appendix A). Such approaches are also the most straightforward. They include the direct addition of an autologous-derived TAA alone, a combination of them, or the whole tumor cell lysate to the culture medium with DCs and incubation with them, before or after DC maturation [192]. Further, proteins that entered DCs will be involved in processing and cross-presentation. However, such approaches encounter several problems. As tumor cells are required to be autologous, a low yield of the patient’s tumor tissue could be a limiting factor. A large tumor tissue volume is required in order to harvest TAAs in sufficient amounts [218]. Dendritic cell pulsing with recombinant proteins and peptides could overcome this problem. However, another limiting factor in the case of peptide application arises. To construct peptides that will correspond to the epitopes of chosen TSAs, the patient’s haplotype should be considered. Accordingly, the peptide loading approach becomes complicated due to the need to screen and select epitopes that would bind a patient’s MHC molecules. Another problem is that the nature of the immune response generated by DCs depends heavily upon the mode of antigen uptake [219,220]. Straightforward pulsing of DCs is inferior in comparison to the targeting of antigens to specific receptors of DCs. Antigens conjugated with receptor-specific antibodies or antigen modulation for specific recognition by DC receptors enhance antigen uptake and they are more likely to undergo cross-presentation. The same strategies for DC targeting in vivo discussed above could be applied in the case of ex vivo DC loading since all proposed in vivo DC-targeting vaccines undergo in vitro studies with in vitro cultures of DCs. Thus, the targeting of DC receptors, but ex vivo, could also enhance the processing and cross-presentation efficacy of DCs.

### 3.3. Dendritic Cell Transduction with Antigen-Encoding mRNA, DNA

Compared to TAA or TSA loading, this approach is an attractive option due to the possibility of avoiding the need for identification of the patient’s haplotype, as well as to avoid the requirement for antigen harvesting or production. It has been demonstrated that the transfection of mRNA encoding TSAs into DCs can induce an antigen-specific CD8^+^ and CD4^+^ T cell response [221,222]. The following step of artificial DC maturation is required. Although this approach has been demonstrated to elicit a CTL response, it is limited due to low transfection efficacy. Lipid-mediated mRNA transfection was proposed to enhance transfection efficacy. Nevertheless, it has been demonstrated that lipid-mediated mRNA transfection was not substantially effective compared to passive mRNA transfection [221]. Moreover, this approach should be applied providently due to the potential that the lipids could be quite toxic. Electroporation has been shown to be the most effective method of mRNA transfection [195,223]. This approach is preferable due to the lack of a need to add additional substances and its higher efficacy compared with passive and lipid-mediated approaches. Electroporation of DCs has been successfully used in preclinical and clinical trials for treating cancer [224,225]. Recent advances in the mRNA transfection approach are related to the so-called TriMix-formula. This approach implies mRNA transfection-based delivery of antigens alongside with CD40L, CD70 and TLR4 [226]. Received TriMix-DCs demonstrate an enhanced T cell activation potential [227,228]. Vaccination with autologous TriMix-DCs has been shown to be safe and capable of antigen-specific immune response activation [229].

Antigen-encoding DNA delivery to DCs has been also applied. Recently, several nanoparticle-based approaches to DNA delivery have been reported. Liposomes or gold nanoparticles functionalized with mannose-mimicking headgroups were used to deliver DNA plasmid to DCs ex vivo [81,230]. Although this approach demonstrates some efficacy, further study is required for translation to clinical studies.

### 3.4. Dendritic Cell Transduction with Viruses

An alternative approach to ex vivo DC loading is transfection with viral vectors that encode TSAs. It is possible to construct recombinant viruses that will encode TSAs, while deleting out genes encoding virulence or replication factors. The advantage of such an approach is that the virus will initiate natural DC activation, therefore enhancing the immunogenicity of the encoded TSAs [231]. In addition, in some cases, the viral vector itself can act as a DC maturation signal. This option allows the procedure of DC exposure to additional cytokines to achieve DC maturation to be avoided. Another advantage is the addition of cytokine and co-stimulatory molecule encoding genes that will result in enhanced DC immunogenicity. It should also be noted that the viral vector approach can be used for in vivo DC targeting. However, preexisting immunity against the vector will reduce the ability to induce an in vivo response [202]. Several viruses are currently utilized as DC-targeting vectors, with lentiviruses the most commonly used [232,233]. Lentivirus-based vectors have several advantages over other currently used viral vectors in clinical trials. Numerous studies have confirmed the stable transduction of moDCs with the stable expression of genes encoded by the lentivirus [194,234]. As for vector-dependent immunogenicity, it has been demonstrated that lentivirus-based vectors elicit a strong tumor-specific CD8^+^ and CD4^+^ T cell response [235]. Such DCs pulsed with a lentivirus-based vector were able to inhibit the growth of preexisting tumors [236]. Moreover, another advantage of lentivirus-based vectors has been described, in that they can transduce quiescent and non-dividing cells, which is especially applicable as moDCs are usually propagated from quiescent CD14^+^ or CD34^+^ progenitors [231]. Modifying the lentiviral-based approach, so-called “smart DCs”, with the ability to self-differentiate and present antigens on MHC molecules, have been developed [237]. In this approach, monocytes were transduced with lentiviral-vector encoded GM-CSF and IL-4, together with TSA.

### 3.5. Dendritic Cell Maturation/Stimulation Approaches

After iDCs are loaded or transduced in respect to the chosen method, they should receive activation and maturation stimuli. The biology and reason for this step are described in detail in Section 2.5 of this review. The desired outcome is to induce a high expression of MHC I and II; co-stimulatory molecules CD40, CD80, CD83 and CD86; secretion of Th1 inflammatory cytokines IL-12 and IFNs, and the expression of chemokines such as CCR7 to polarize DCs towards Th1 activation [208]. Much the same pathways and receptor agonists as described above are used for artificial ex vivo DC maturation and activation, which include TLR and CD40 ligands and prostaglandin E2 (PGE_2_). However, since DC differentiation will take place ex vivo, additional cytokines, such as TNF-α, IL-1β, IL-6, IFN-α and IFN-γ, are also required to imitate the natural cell microenvironment. The most typical “maturation cocktail” to which DCs are exposed consists of TNF-α, IL-1β, IL-6 and PGE_2_ [238]. However, the composition of such “maturational cocktails” varies from study to study.

Another approach to initiate DC maturation is to genetically modify iDCs to express co-stimulatory molecules. A recent advance is to use mRNA transfection-based delivery of CD40L, CD70 and constitutively active TLR4. Such a formula was designated as “TriMix”. Besides effective maturation, TriMix-transfected DCs have been shown to decrease Treg cell-mediated suppression of DCs and CD8^+^ T cells, as well as to partially lose their own suppressive capability [228]. These results were accompanied by a decrease in CD27 and CD25 expression on Treg cells, as well as in increase in of IFN-γ, TNF-α and IL-10 expression, suggesting a shift from a Treg phenotype to a Th1 phenotype [228]. As described above, the TriMix formula has been reported to accompany antigen-encoding mRNA transfection. Thus, it is possible to perform a single-step loading and maturation of DCs by the transfection of TSA-encoding mRNA alongside TriMix, which simplifies ex vivo DC handling [227].

### 3.6. Dendritic/Tumor Fusion Cells

The most recently proposed strategy of ex vivo DC loading is manufacturing DC/tumor FCs. The essence of this approach, similar to hybridomas, is the generation of the hybrid cells through the fusion of autologous-derived DCs and tumor cells (Figure 3). Hybrids obtained in such way are designated “dendritoma”. Tumor-specific peptides and TSAs have been widely used for DC vaccination, but this approach suffers from the drawback of a limited number of known MHC-restricted tumor peptides. In addition, monoclonal peptide-specific CD8^+^ CTLs may not be sufficiently effective to treat cancer patients [239]. The development of DC/tumor FCs appears to be a fascinating approach to overcome these problems [240]. Dendritic/tumor FC vaccines possess the necessary components for the processing and presentation of a tumor antigen to the host immune cell to induce an effective anti-tumor immune response. The fusion of DCs and whole tumor cells by chemical, physical, or biological means creates heterokaryon. This heterokaryon possesses DC-derived MHC class I, MHC class II, DC-derived co-stimulatory molecules (lymphocyte function-associated antigen 1 and 3, CD-40, ICAM-1) and tumor-derived antigens and all of these molecules represent efficient machinery for antigen processing and cross-presentation [241]. In DC/tumor FCs, the cytoplasm of both DCs and whole tumor cells is integrated without nuclear fusion, as demonstrated by immunoelectron microscopy [242,243]. These morphological features allow the retention of the functions of both original cell types, including co-expression of tumor-derived whole TAAs (both known and unidentified) and DC-derived MHC class I and II molecules [240,243]. In general, DC/tumor FCs process multiple antigenic peptides from whole tumor cells and load them onto MHC class I molecules in the endoplasmic reticulum. The antigenic peptide-MHC class I complexes are expressed on the DC/tumor FC surface and presented to CD8^+^ T cells. The endogenous pathway of direct antigen processing and presentation in DC/tumor FCs is preserved. Dendritic/tumor FCs can also synthesize MHC class II-restricted antigenic peptides from whole tumor cells in the endoplasmic reticulum. Dendritic cell-derived MHC class II molecules and tumor-derived antigenic peptides travel by separate routes and converge to form MHC class II-peptide complexes in DC/tumor FCs, where MHC class II-antigenic peptide complexes are expressed on the DC/tumor FC surface and presented to CD4^+^ T cells [244]. Therefore, polyclonal antigen-specific CD4^+^ and CD8^+^ T cells are directly induced by DC-tumor FCs in the draining lymph node [245].

In summary, the DC/tumor FCs offer the following advantages for the induction of anti-tumor immune responses: DC/tumor FCs cross-present whole tumor-derived antigenic peptides, which avoids the need to identify antigenic peptides for individual patients; a broad array of known and unidentified TAAs can be simultaneously presented on the surface of DC/tumor FCs, which increases the frequency of polyclonal antigen-specific CD4^+^ and CD8^+^ T cells, resulting in long-term efficient anti-tumor immunity; numerous TAAs are presented in the context of co-stimulatory molecules, which prevents tolerance induction, resulting in an efficient anti-tumor immune response; and DC/tumor FCs migrate into draining lymph nodes and form clusters with CD4^+^ and CD8^+^ T cells in the T cell area of lymph nodes, such that DC/tumor FCs do not have to take up exogenous TAAs to activate CD4^+^ and CD8^+^ T cells.

Koido and coworkers successfully transfected MUC1 to murine MC38 adenocarcinoma cells which were fused to synergistic DCs derived from bone marrow in the presence of polyethylene glycol (PEG) [246]. Tumor regression was observed in the mice following administration of the fusion vaccine against MUC1-positive tumor cells. With this experiment, it can be concluded that FCs can induce CTLs against MUC1 and thus reverse the tolerance to human MUC1 antigens [246]. For effective cancer vaccination, it is necessary to produce long-lasting CD4^+^ and CD8^+^ T cell responses and CTL responses. It is now well documented from the literature that this can be achieved by DC/tumor cell fusion [242,247,248]. In addition, DC/tumor FCs could be efficiently frozen to retain the potency of antigen presentation or the capacity to stimulate T cells to induce CTL responses. The cryopreserved DC/tumor FCs have potential applicability in the field of anticancer immunotherapy and provide a platform for adoptive immunotherapy in the clinical setting. Dendritic/tumor FC vaccines are in the initial phase I/II of clinical trials. Krause et al. performed an experiment in which allogeneic DCs were fused with autologous tumor cells and given subcutaneously to a group of patients, wherein 44% of patients responded positively to this fusion vaccine [249]. Similar results were reported by Haenssle and colleagues [249,250].

## 4. Conclusions and Outlook

Cancer immunotherapy development is experiencing its golden age. The current methodology in molecular biology provides an opportunity to deeply examine the interaction of malignant cells with immune system cells, as well as to interfere with them for cancer treatment. As for DCs, our understanding of their nature has been much improved since their discovery by Ralph Steinman and his colleagues, including the DC-based vaccination field. Enormous numbers of strategies that describe the employment of DCs in cancer immunotherapy are being proposed. Further, many clinical trials have been conducted that have confirmed the potentially high anticancer efficacy and the safety of DC-based vaccines. However, there is a problem of result comparison between trials which used different DC loading or targeting approaches. Even the same antigen delivery approach could be different from trial to trial due to differences in the DCs generation method used, the applied methods of DC maturation, the selection of antigens and administration routes. Accordingly, it is quite difficult to reach a conclusion about the most preferable antigen delivery approach, as well as to make a decision regarding which approach to apply in one case or another. To overcome this, the development of more standardized trial designs and treatment settings that will reduce alternatives is necessary. To enable conclusions to be drawn and to compare results from different trials, more standardized, accurate and detailed monitoring with clear outcome assessment criteria are required. However, even immune response assessment differs from trial to trial (Appendix A). Moreover, most of the clinical efficacy data rely on short-term criteria, while there are very few completed phase III trials (Appendix A) [191]. There is an urgent need for clinical studies demonstrating whether DC-based vaccines can induce a durable response and improve long-term survival.

Although the standardization of DC-based vaccination is required for comparable outcome characterization, the most superior DC-based approach remains to be determined. Only a few studies have directly compared DC-based loading or targeting approaches with each other in humans (Table 3). Dendritic/tumor cell hybrids and DC-tumor cell mixtures have been compared in an in vitro study showing electroporation-mediated fusion of moDCs with tumor cells to be superior for T cell activation [251]. Another clinical trial in melanoma compared an moDCs-tumor cell co-culture, PEG-mediated moDC/tumor FCs and tumor lysate-loading of moDCs [252]. This trial indicated that a mixture of tumor cells and moDCs may be inferior to the fusion and lysate loading (however, small numbers precluded a significant correlation). Although DC/tumor FCs showed better results, more studies comparing other techniques to determine the optimal approach for ex vivo antigen loading of DCs have yet to be performed. Attempts to compare in vivo DC targeting and ex vivo DC loading have also been performed, with the viral-based approach in particular. One study reported a comparison of viral-based in vivo DC targeting and ex vivo DC loading [253]. The example of the so-called PANVAC vaccine, which is a vaccinia virus encoded with CEA and MUC-1 antigens together with B7.1, ICAM-1 and LFA-3 co-stimulatory molecules, has demonstrated that in vivo viral targeting of DCs is more efficient than the administration of autologous moDCs transduced with PANVAC artificially [253]. Recurrence-free survival was 28.9 and 22.9 months, respectively [253].

Both in vivo DC targeting and ex vivo DC loading have advantages and disadvantages (Table 4). However, cancer is a heterogeneous disease. Frequently, one approach could result in completely different outcomes in patients with different nosology, or even in patients with the same nosology. Therefore, the next generation of DC-based vaccines should be optimized for selective use in individual patients based on their tumor biology. Examples of vaccine selection criteria could be the following: the inability to harvest enough tumor material; the inability to harvest enough DC precursors; the presence of highly heterogeneous tumor with TAAs and TSAs less than fully described. Further, in accordance with the identified criteria, one of the in vivo DC-targeting approaches, to prevent the lack of tumor material or cells, or ex vivo DC loading approaches, to prevent gaps in TAAs and TSAs identification, could be applied. Although the ex vivo DC loading approach is continuing to develop, with the latest advances as an application of HSP-TAAs complexes derived from DC/tumor FCs, or MHCs-epitopes bearing extracellular vesicles derived from ex vivo loaded DCs [257,258], in our subjective opinion, it makes sense to expect a greater development of the in vivo DC-targeting approach. This approach is consistent with the current trend of personalized and precision approaches in medicine. Recently, a new field in medicine—theranostics—based on such trends has emerged [259]. Theranostics implies a personalized approach, endeavoring to apply less invasive procedures, early diagnosis and specific targeted therapy based on specific diagnostic tests. We believe that in vivo DC-targeting vaccines prospectively fit this landscape, comparing their pros and cons with the ex vivo loading approach (Table 3). The key to theranostics is a sequential approach when the therapeutic target is identified according to the results of specific diagnostic tests. Thus, the identification of the tumor transcriptome and proteome to identify more immunogenic TSAs is helpful. Several highly immunogenic TSAs have already been described [260,261,262,263]. However, modern methods such as high-throughput sequencing technologies and bioinformatics analysis of big data allow the identification of new tumor antigenic determinants, such as neoantigens [264,265]. This makes it possible to analyze and determine the TSAs specifically expressed by each cancer patient’s tumor, with further TSA or recombinant production and utilization of its peptide in the vaccines that target DCs in vivo. Flexible platforms that represent the targeting vector with self-adjuvant properties will be required as a basis to enable the possibility of personalized vaccines. Examples of such platforms, such as Clec9A-specific nanoemulsions, Lipovaxin and Polyman26, already exist [46,90,131]. Thus, only the substitution of necessary antigens will be required.

It has become clear that there is no single universal pill to cure every cancer. To cure cancer, an integrated approach should be applied. The reactivation of an immune cell response alone in cancer is not enough. Although there are certain cases of patients on DC-based therapy who fully responded in each trial, DC-based vaccines have failed to achieve more than 15% objective response rates in cancer patients [266]. This is explained by the immunosuppression in the tumor microenvironment. Accordingly, DCs and T cells that infiltrate tumors or which fall into the tumor microenvironment become tolerogenic, even when the DC-based vaccines could successfully generate tumor-specific peripheral CTLs [191]. This problem could be overcome by the combination of DC-based vaccination with ICI therapy—a currently investigated and effective method approved by the FDA to inhibit tumor-infiltrating immune cells’ immunosuppressive signaling [267]. Such a combinatorial approach could allow reactivated DCs to activate the CTL response in the tumor microenvironment. Moreover, numerous other recent immunotherapeutic strategies in cancer have been developed, such as adoptive cell transfer therapy, a chimeric antigen receptor-based approach and neoadjuvant cytokine treatment [268,269]. The simultaneous utilization of complementary approaches would enhance the clinical response by covering the deficiencies of each alone. Clinical trials are already underway to assess DC-based vaccination within a landscape of such combinatorial approaches to cancer treatment [32].

## Figures and Tables

**Figure 1 cancers-12-00590-f001:**
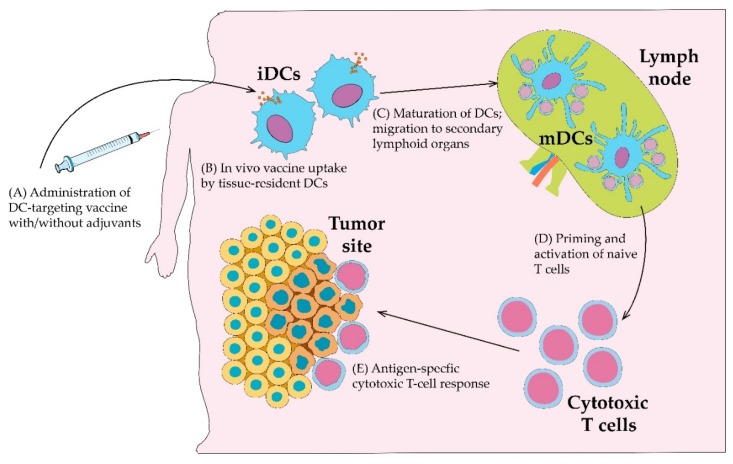
Representation of in vivo dendritic cell (DC) targeting strategies. (**A**) Administration of chemically modified tumor proteins or peptides; proteins or peptides fused or in complex with antibodies or other DC-targeting vectors; with/without adjuvants. Different administration routes can be utilized depending on the purpose and the DC subset they are required to reach: subcutaneous, intradermal, intranodal, intralymphatic and intravenous. (**B**) After administration into a DC-rich site of the organism, the antigen-containing vaccine is recognized by immature DCs (iDCs), followed by its internalization. (**C**) Subsequently, iDCs undergo the maturation process, which includes a decrease in antigen-capture activity, migration into lymph nodes and an increase in MHC and co-stimulatory molecule expression. (**D**) After the maturation process, mature DCs (mDCs) are capable of antigen presentation to naive T cells, involving the presentation of processed peptides to T cells via MHC-TCR interaction. (**E**) Finally, antigen-specific cytotoxic T lymphocytes (CTLs), activated by mDCs, are capable of recognizing tumor cells, followed by a cytotoxic response.

**Figure 2 cancers-12-00590-f002:**
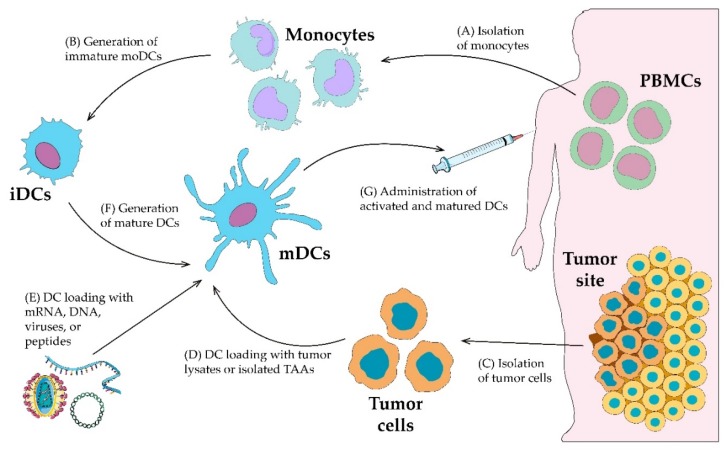
Representation of ex vivo DC vaccine generation. (**A**) Isolation of monocytes for further differentiation into immature moDCs could be performed from the peripheral blood of the patient. (**B**) Differentiation of monocytes into immature moDCs takes five to seven days in the presence of GM-CSF and IL-4. Further steps of DC loading and maturation are dependent on the method. For DC loading with tumor lysate or tumor-derived proteins and peptides (**D**), the patient’s tumor cells are isolated from the tumor site (**C**). On the other hand, iDCs could be loaded with mRNA, viruses, or in vitro prepared peptides (**E**). Whatever method of ex vivo DC loading is chosen, iDCs are exposed to the “maturation cocktail” simultaneously with antigen loading, which typically consists of TNF-α, IL-1β, IL-6 and PGE2 (**F**). Like the administration of DC-targeting vaccines, DC vaccines generated ex vivo could be administered into different sites of the patient’s body (**G**).

**Figure 3 cancers-12-00590-f003:**
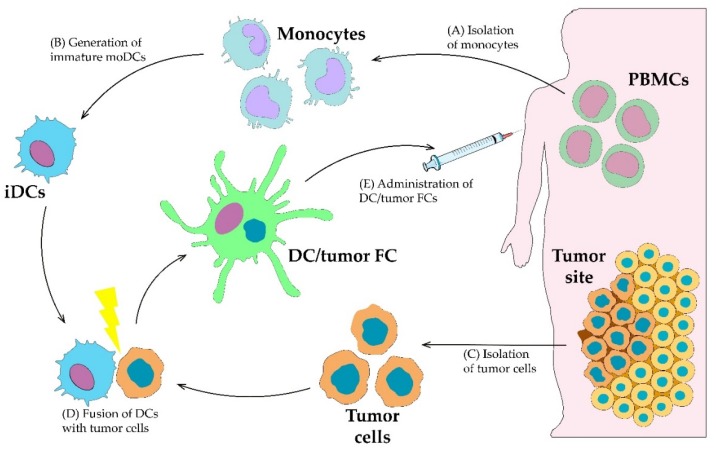
Representation of DC/tumor fusion cells (FC) vaccine generation. (**A**) Isolation of monocytes from the patient’s peripheral blood. (**B**) Generation of monocyte-derived DCs (moDCs) is presented in the same way as for ex vivo DC loading with antigen. After isolation of autologous (or allogenic) tumor cells (**C**), whole cells are mixed with previously prepared moDCs and their fusion is performed by different chemical or physical methods (for example PEG-mediated or electroporation-mediated methods) (**D**,**E**) Administration of received DC/tumor FCs is performed as previously described into different sites of the patient’s body.

**Table 1 cancers-12-00590-t001:** Clinical trials of in vivo DC (antigen-presenting cells (APCs)) targeting in cancer with published results.

DC (APC) Targeting Method	Condition	Trial Phase	Trial ID	Refs
MR targeting	Different advanced cancers	I	NCT00709462; NCT00648102	[44]
DC-SIGN targeting	Different NY-ESO-1-expressing tumors	I	NCT02122861	[45]
DC-SIGN targeting	Melanoma	I	N/A	[46]
DEC-205 targeting	Different advanced cancers	I	NCT00948961	[47]
SR targeting with HSP	Pancreatic adenocarcinoma	I	N/A	[48]
SR targeting with HSP	Glioblastoma	I	NCT02122822; ChiCTR-ONC-13003309	[49]
SR targeting with HSP	Glioblastoma	II	NCT00905060	[50]
SR targeting with HSP	Glioblastoma	II	NCT00293423	[51]
SR targeting with HSP	Cervical intraepithelial neoplasia III	II	NCT00075569	[52,53]
SR targeting with HSP	Melanoma	II	N/A	[54]
SLOs DCs targeting	Melanoma	II	N/A	[55]

N/A—not available; SLOs—secondary lymphoid organs.

**Table 2 cancers-12-00590-t002:** Expression of various CLRs on human DC subsets [41,60,61].

DC Subset	DEC205/CD205	MR/CD206	Langerin/CD207	DC-SIGN/CD209	BDCA2/CD303	Clec4A/DCIR	Clec9A/DNGR1	Clec12A
moDC ^1^	+	+	−	+	−	+	−	+
Blood-resident DCs								
cDC1	+	−	−	−	−	+	+	+
cDC2	+	−	−	−	−	+	−	+
CD16^+^ DC ^2^	+	−	−	−	−	+	−	?
pDC	+	−	−	−	+	+	−	+
Tissue-resident DCs								
cDC1	?	−	−	−	−	+	+	?
cDC2	−	? ^3^	± ^4^	−	−	?	−	?
LCs ^5^	+	−	+	−	−	?	?	?
IDECs	?	+ ^6^	−	+^6^	?	?	?	?

^1^ moDC represent monocyte-derived DCs cultured ex vivo and are given for comparison. ^2^ This subset of cells has instead been attributed to CD16^+^ non-classical monocytes by recent studies [62,63]. However, further studies are required to identify the phenotypic and functional nuances of this subtype. ^3^ Whereas there are studies which reported mannose receptor (MR) expression on dermal cDC2, other studies reported that these are macrophages, but not dermal DCs, which express MR among “dendritic-appearing” cells in derma [64,65]. ^4^ Langerin^+^ and langerin^−^ cDC2 subsets have been described [66]. ^5^ It is still heavily debated whether Langerhans cells (LCs) should be classified as DCs or macrophages. ^6^ It has been shown that inflammatory dendritic epidermal cells (IDECs) express MR [67]. However, the IDEC subpopulation is still poorly described, thus, MR and dendritic cell-specific intercellular adhesion molecule-3-grabbing non-integrin (DC-SIGN) expression on DCs is still under discussion. (?)—unknown status.

**Table 3 cancers-12-00590-t003:** Comparative studies of different DC targeting/loading approaches in humans.

Type of Study	Type of Vaccine		Combinatorial Treatment	Detected Response	Refs
	DC Target/Source	Targeting/Loading Method			
In vitro	cDC2 or moDCs ex vivo targeting	Anti-DEC205 targeting	LPS	CD4^+^ and CD8^+^ T cells	[254]
		Anti-MR targeting			
		Anti-CD40 targeting			
In vitro	pDC ex vivo targeting	Anti-DEC205 NPs	Each NP encapsulate R848 TLR7 agonist	CD4^+^ and CD8^+^ T cells	[255]
		Anti-DCIR NPs			
		Anti-BDCA2 NPs			
		Anti- FcγRIIa NPs			
In vitro	moDCs ex vivo loading	Coculture of moDCs with MDA-MB-231 or KA2 cells		CD8^+^ T cells	[251]
	moDCs ex vivo fusion	Electroporation-mediated fusion of moDCs with MDA-MB-231 or KA2 cells		CD8^+^ T cells	
Phase II clinical trial	In vivo targeting	Administration of poxvector	GM-CSF	T-cell response	[253]
	moDCs ex vivo loading	Loading with poxvector			
Phase II clinical trial	In vivo targeting	Administration of autologous tumor-derived cells		N/A	[256]
	moDCs ex vivo loading	Coculture with autologous tumor-derived cells			
Phase II clinical trial	moDCs ex vivo loading	Coculture with autologous tumor cells		CD8^+^ T cells; DTH-test	[252]
	moDCs ex vivo fusion	PEG-mediated fusion with autologous tumor cells		CD8^+^ T cells; DTH-test	
	moDCs ex vivo loading	Pulsing with autologous tumor cell lysate		CD8^+^ T cells; DTH-test	

N/A—was not assessed.

**Table 4 cancers-12-00590-t004:** Pros and cons of in vivo targeting versus ex vivo loading (Tacken et al., 2007, with modifications) [95].

Pros/Cons	In Vivo Targeting	Ex Vivo Loading
Pros	Off the shelf use: Lower costs at large-scale production One specialized GMP manufacturer One procedure for product control Equal product quality among clinical centers Accessible to a large number of patients Clinical interventions limited to vaccinations Optimal antigen delivery within the natural environment: Antigens can be targeted either to multiple, or to a single DC subset by targeting one or multiple receptors DCs, even cDCs, are reached and activated within the natural environment and at multiple sites	Highly controlled maturation and activation: DCs can be properly stimulated ex vivo and maturation status is checked before administration High DC specificity: Only the ex vivo cultured DCs are reached Wide TAAs specificity: Loading and thus cross-presentation of multiple TAAs could be achieved
Cons	Poor control of maturation and activation: DCs activated and matured in vivo, stimuli (adjuvant) need to be administered systemically, incorporated into the targeting vector, or targeted by itself Limited specificity: Most receptors are not specific for a single cell type Only few receptors, utilized for DC targeting, are true DC-specific	Tailor made: Labor-intensive procedure for each individual patient High costs, mainly independent of the number of procedures Multiple procedures for product control at different sites Product quality differs per production site, procedure and patient Accessible to a limited number of patients Requires cytapheresis Limitations to DC subsets and in vivo distribution: Limited to DC subsets that can be isolated in sufficient numbers or cultured in vitro Poor distribution of DCs injected at high concentrations at specific sites

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
