# Peer review of "Dendritic Cells in Anticancer Vaccination: Rationale for Ex Vivo Loading or In Vivo Targeting"

_cancers, 2020, doi:10.3390/cancers12030590_

Round 1

Reviewer 1 Report

In this review, Baldin and colleagues focused on the methods of current DC based vaccine and the strategies that applied to enhance the efficacy. The review is well written and this reviewer has a few suggestions which may further improve this exciting work. 1, Page 2, line 80. Antigen processing can be briefly discussed in this review and related references should be included as this is critical for the DC function. 2, Page 3,line 105-106. The authors may want to explain immune checkpoint a bit more here, before discussing its inhibitors. Line 116. "IL-2 with a prolonged half-life" looks strange. The authors may want to explain it with the original references. 6, Page 5, line 199. It will be great if the authors can select or combine references of those related clinical trials into a small table and place it as Table 1. 7, Page 25, line 1116-1120. The authors may want to reshape their phrase organization to highlight the criteria.

Author Response

“In this review, Baldin and colleagues focused on the methods of current DC based vaccine and the strategies that applied to enhance the efficacy. The review is well written and this reviewer has a few suggestions which may further improve this exciting work.”

“Page 2, line 80. Antigen processing can be briefly discussed in this review and related references should be included as this is critical for the DC function.”

Our response:

A brief discussion of antigen processing with related references was added to the Introduction part in the main text (line 78-99) as requested by the reviewer.

“Page 3, line 105-106. The authors may want to explain immune checkpoint a bit more here, before discussing its inhibitors.”

Our response:

We would like to point attention on the fact that immune checkpoints are discussed before mentioned lines, briefly starting with initial sentences about immune tolerance that can be promoted by DCs and flowing into the discussion of the importance of PD1 and CTLA4 interaction with their ligands in this process (lines 108-119). Nonetheless, additional explanation and summary of immune checkpoints were added (lines 119-122).

“Line 116. "IL-2 with a prolonged half-life" looks strange. The authors may want to explain it with the original references.”

Our response:

“IL-2 with a prolonged half-life” means that IL-2 molecule was modified in such a way that it allows increasing the time until half of administered IL-2 will be eliminated or decayed. Generally, this allows longer circulation of IL-2. Modification (fusion) of IL-2 with mouse serum albumin was used in the referenced study. A small description of what “IL-2 with a prolonged half-life” is was added to the corresponding sentence in brackets (line 137). Reference to the original study was added (line 137).

“Page 5, line 199. It will be great if the authors can select or combine references of those related clinical trials into a small table and place it as Table 1.”

Our response:

Related references describing clinical trials of in vivo DC targeting have been presented in the “In vivo targeting” section of Table S1 from Supplementary Materials to the manuscript. Additionally, Table 1 that includes requested references was added to the main text after the mentioned line (lines 223-224).

“Page 25, line 1116-1120. The authors may want to reshape their phrase organization to highlight the criteria.”

Our response:

The mentioned sentence was restructured in two sentences. Selection criteria were highlighted in the first sentence and an explanation of how to possibly apply them was described in the second (lines 1141-1146).

Reviewer 2 Report

This manuscript titled by Dendritic Cells in Anticancer Vaccination: Rationale for Ex Vivo Loading or In Vivo Targeting broadly summarized the latest applications about dendritic cells related vaccination and immunotherapy. Actually, extracellular vesicles (EV), including exosomes and microvesicles have been widely studied in immunotherapy in recent years. The authors need to incorporate some of the researches and discuss their potential utilization in antigen-presenting and tumor immunotherapy.

Author Response

“This manuscript titled by Dendritic Cells in Anticancer Vaccination: Rationale for Ex Vivo Loading or In Vivo Targeting broadly summarized the latest applications about dendritic cells related vaccination and immunotherapy.”

“Actually, extracellular vesicles (EV), including exosomes and microvesicles have been widely studied in immunotherapy in recent years. The authors need to incorporate some of the researches and discuss their potential utilization in antigen-presenting and tumor immunotherapy.”

Our response:

Extracellular vesicles (EVs) are presenting quite an interesting approach to cancer immunotherapy. However, after studying the mentioned issue, in our opinion, cancer immunotherapy with EVs is not relevant to this review. The topic of the present review (also according to its headline) is different methods of vaccines (antigens) delivery to dendritic cells (DCs) using in vivo DC targeting, or ex vivo DC loading approaches. The proposed for discussion EVs for immunotherapy are used as an agent that mimics DCs. Being isolated from the same DCs, they are already the substance that presents the antigen in complex with MHC I or MHC II to the corresponding T cells. However, the delivery of antigens to DCs, from which EVs are isolated subsequently, is performed by one of the methods described in the present review. Thus, in the context of EVs and cancer immunotherapy, there is no question about their utilization as one of the strategies of antigen delivery to DCs.

Nonetheless, a brief reference of the use of EVs isolated from DCs preloaded with tumor antigens/epitopes was added to the Conclusion and Outlook section of the main text as requested by the reviewer (lines 1146-1149).